# Stress-dependent conformational changes of artemin: Effects of heat and oxidant

**Zeinab Takalloo**[1☯], **Zahra Afshar Ardakani**[1☯]**, Bahman Maroufi**[2]**, S. Shirin Shahangian**[3]**, Reza H. Sajedi**[1]*

**1** Department of Biochemistry, Faculty of Biological Sciences, Tarbiat Modares University, Tehran, Iran, **2** Mah Behin Gene Gostaran Company, Tehran, Iran, **3** Department of Biology, Faculty of Sciences, University of Guilan, Rasht, Iran

☯ These authors contributed equally to this work.
* sajedi_r@modares.ac.ir

**Data Availability Statement:** We confirm that the data provided in our Supporting Information as the "minimal data set", reach the conclusions drawn in the manuscript with related metadata and

## Abstract

Artemin is an abundant thermostable protein in *Artemia* embryos and it is considered as a highly efficient molecular chaperone against extreme environmental stress conditions. The conformational dynamics of artemin have been suggested to play a critical role in its biological functions. In this study, we have investigated the conformational and functional changes of artemin under heat and oxidative stresses to identify the relationship between its structure and function. The tertiary and quaternary structures of artemin were evaluated by fluorescence measurements, protein cross-linking analysis, and dynamic light scattering. Based on the structural analysis, artemin showed irreversible substantial conformational lability in responses to heat and oxidant, which was mainly mediated through the hydrophobic interactions and dimerization of the chaperone. In addition, the chaperone-like activity of heated and oxidized artemin was examined using lysozyme refolding assay and the results showed that although both factors, i.e. heat and oxidant, at specific levels improved artemin potency, simultaneous incubation with both stressors significantly triggered the chaperone activation. Moreover, the heat-induced dimerization of artemin was found to be the most critical factor for its activation. It was suggested that oxidation presumably acts through stabilizing the dimer structures of artemin through formation of disulfide bridges between the subunits and strengthens its chaperoning efficacy. Accordingly, it is proposed that artemin probably exists in a monomer–oligomer equilibrium in *Artemia* cysts and environmental stresses and intracellular portion of protein substrates may shift the equilibrium towards the active dimer forms of the chaperone.

## Introduction

The brine shrimp *Artemia* is a micro-crustacean, well adapted to the extreme conditions such as desiccation, radiation, high and low temperatures and several years of anoxia. Populations of *Artemia* are found in many inland salt lakes and coastal salterns distributed all over the world [1]. This organism undergoes diapause by forming a unique structure called cyst

methods, and any additional data required to replicate the reported study findings. In addition, we also provided the original uncropped and unadjusted gel images as "S1_raw_images" file in the revision. We would like to provide contact information of three involved authours for ensuring data access. The email contacts include: Zeinab Takalloo (First Author): z.takalu@yahoo.com S. Shirin Shahangian (Professor Assistant & Project Advisor): shahangian@guilan.ac.ir Reza H. Sajedi (Professor of Biochemistry & Project Supervisor): sajedi_r@modares.ac.ir.

**Funding:** MBGG (Mah Behin Gene Gostaran Company) provided some instruments for authors, and contributed in analysis of spectroscopic data.

**Competing interests:** MBGG provided some instruments for authors, and contributed in analysis of spectroscopic data. The commercial affiliation does not alter our adherence to PLOS ONE policies on sharing data and materials.

enabling it to survive in the adverse ambient conditions, which can remain viable for decades [2]. Artemin is a stress protein of encysted *Artemia* embryos, representing about 10% of the soluble cellular proteins, but it is almost completely absent from nauplius larvae [3]. Due to its high structural stability, abundance and chaperone-like activity, artemin probably contributes to cyst stress resistance [4]. Artemin monomers consist of 229 amino acid residues and exhibit a molecular mass of 26 kDa, which are assembled into rosette-like oligomers of ~600–700 kDa consist of 24 monomer subunits [4, 5]. Artemin and ferritin, an iron storage protein, show a degree of homology in both sequence and structure, although artemin contains 45–50 extra residues compared to ferritin [3, 5–7]. In contrast, artemin fails to bind iron due to the extension of carboxyl terminal of the protein monomers, which suggests to have a role in its chaperone function [5, 8]. It is very heat stable and associates with RNA at elevated temperatures, suggesting protection of RNA [9].

Previous researches revealed a chaperone-like activity for artemin, responsible for protecting transfected cells and preventing protein aggregation *in vitro* [10]. Our previous computational and experimental studies on artemin from *Artemia urmiana* (Urmia Lake, Iran) also confirmed the potency of artemin in suppressing protein aggregation [5, 8, 11]. Besides, artemin was capable to protect proteins and cells against different stress conditions such as oxidant, cold and salt [12, 13]. These studies proposed that artemin binds to protein substrates *via* hydrophobic interactions. Artemin is a histidine/cysteine-rich protein and due to the high content of cysteines and their distributions, it was suggested that the oxidative state of cysteines modulates the redox-regulated activity of artemin [3, 12]. Therefore, in our recent study, the modification of cysteine residues revealed that the function of artemin is greatly dependent on the formation of intermolecular disulfide bonds suggesting artemin as a redox-regulated chaperone [14].

To date, artemin has been used for different purposes. For example, it could enhance the soluble production of some aggregation-prone proteins in bacterial cells such as aequorin [15] and luciferase [13]. Besides, the anti-amyloidogenic effect of artemin on α-synuclein [16] and blocking fibrillization of tau protein [17] were further confirmed. Despite such successes, there are no enough details on the protein activation mechanisms or structural modifications of this chaperone. Our suggestion is that conformational changes of artemin are responsible for its chaperone function upon exposing to stress conditions and interactions with the protein substrates. In fact, these modifications may change the surface hydrophobicity of the chaperone, which result in hydrophobic interactions with target proteins. Accordingly, in the present study, we tried to provide some basis on the conformational changes of artemin upon exposing to stress conditions and interaction with the protein substrates. The tertiary and quaternary structures of artemin under heat and oxidative stresses have been evaluated by intrinsic and extrinsic fluorescence measurements, protein cross-linking analysis, and dynamic light scattering (DLS). In addition, the activity of artemin was investigated under both stress conditions using dilution-induced aggregation assay of lysozyme. Notably, this is the first report in providing details on the conformational changes of artemin's subunits. We believe that this information would help us to justify the protective function of artemin in *Artemia* cysts during stress conditions.

## Materials and methods

### Chemicals

Glycine, Tris, NaCl, sodium dodecyl sulfate (SDS), imidazole, kanamycin and Isopropyl-β-D-thiogalactopyranoside (IPTG) were purchased from Bio Basic Inc. (Markham, Ontario, Canada). Peptone and yeast extract were provided by Micromedia Trading House Ltd (Pest,

Hungary). AgNO$_3$ and 8-anilino-l-naph-thalene sulfonic acid (ANS) were purchased from Sigma–Aldrich (St. Louis, MO, USA). Nickel-nitrilotriacetic acid agarose (Ni-NTA agarose) was provided by Qiagen (Hilden, Germany). Chicken egg white lysozyme, dithiothreitol (DTT), glutaraldehyde, H$_2$O$_2$, glycerol, and all other chemicals were obtained from Merck (Darmstadt, Germany). Protein standard marker was purchased from Thermo Fisher Scientific (Waltham, MA, USA).

## Expression and purification of recombinant artemin

pET28a encoding artemin from *Artemia urmiana* was provided [5] and protein expression was carried out in *Escherichia coli* BL21 (DE3) cells as previously described [11]. Purification of the His-tagged protein was carried out using Ni-NTA agarose column. Bound proteins were eluted with a buffer containing 50 mM NaH$_2$PO$_4$, 300 mM NaCl, and 250 mM imidazole, pH 8.0. Aliquots of the eluted protein were taken, followed by dialysis against the phosphate buffer overnight at 4˚C. The protein concentrations were determined using Bradford's method and BSA as standard [18].

## Protein treatments

**Heated artemin (H-artemin).**　Purified artemin (in 50 mM phosphate buffer, pH 7.4) was placed in a heated circulating water bath at temperature range 25–80˚C for 20 min followed by incubation at room temperature (RT) for 20 min before measurements. One standard thermometer was used for taking the temperature of the water in the bath. The temperature of the protein solution in the microtubes was further checked with the thermometer.

**Oxidized artemin (O-artemin).**　Purified artemin (in 50 mM phosphate buffer, pH 7.4) was incubated with 0–160 mM H$_2$O$_2$ for 6 h at 0˚C in dark conditions before measurements.

**Heated oxidized artemin (HO-artemin).**　Purified artemin (in 50 mM phosphate buffer, pH 7.4) was incubated with 0–100 mM H$_2$O$_2$ for 6 h at 0˚C in dark conditions, then placed in a heated circulating water bath at temperature range 25–70˚C for 20 min followed by incubation at RT for 20 min before measurements.

## Conformational analysis

**Intrinsic fluorescence measurements.**　Intrinsic aromatic fluorescence measurements were performed at RT using a LS-55 fluorescence spectrometer (Perkin-Elmer, USA) in a quartz cell of 1-cm path length. Samples containing H-artemin, O-artemin, and HO-artemin in 50 mM phosphate buffer, pH 7.2 were used. The excitation wavelength was set at 280 nm, and the emission spectra were monitored in the wavelength range of 300–400 nm. Excitation and emission bandwidths were 10 nm. Protein concentration for intrinsic fluorescence measurements was about 3.7 μM.

**ANS fluorescence measurements.**　Samples containing H-artemin, O-artemin, and HO-artemin in 50 mM phosphate buffer, pH 7.2 were used. The excitation wavelength was set at 380 nm and emission spectra were taken from 400 to 600 nm at RT by using the fluorescence spectrometer in a quartz cell of 1-cm path length. Excitation and emission slits were set at 10 nm. Protein concentration for ANS measurements was about 9.26 μM and ANS was added at a final concentration of 30 μM.

To investigate whether stress-induced conformational changes of artemin were reversible, the heated and oxidized artemin were kept at 4˚C for 3 h and 48 h, then their intrinsic and extrinsic fluorescence were measured at RT. In the case of oxidized proteins, for elimination of hydrogen peroxide, catalase (8.33 nM) was added in the protein solution, followed by incubating the samples at 4˚C.

**Protein cross-link analysis by SDS-PAGE.** H-artemin, O-artemin, and HO-artemin at final concentration of about 7.4 μM were used. Glutaraldehyde (0.5%) was added to all protein treatments and the reactions terminated after 1 min by addition of 200 mM Tris-HCl, pH 8.0 at the final concentration. In the case of oxidized samples, catalase (8.33 nM) was added to decompose the hydrogen peroxide in the solution before addition of glutaraldehyde. The protein cross-linking were analyzed using 10% non-reducing SDS-PAGE stained with silver nitrate [19]. The non-heated/oxidized artemin (at 25˚C/ 0 mM $H_2O_2$) was used as control. The areas of the bands regarding dimeric states of the protein on SDS-PAGE were calculated in each experiment by densitometry analysis using ImageJ software [20].

In addition, to check whether the dimerization of H-artemin, is dependent on disulfide bridges, DTT (30 mM at final concentration) was added to artemin (3.7 μM), then the samples were incubated at 25 and 60˚C for 20 min followed by incubation at RT for 20 min. Dimerization of the protein was checked using 10% non-reducing SDS-PAGE with and without DTT and glutaraldehyde as the reducing and cross-linking agents, respectively.

**DLS analysis.** The size analysis of the H-artemin (1.85 μM) at 25–80˚C was carried out using dynamic light scattering by Zetasizer Nano ZS instrument (Malvern Instruments Ltd., Malvern, Worcestershire, UK) at RT.

## Chaperone-like activity assay

**Preparation of denatured reduced lysozyme.** Denatured reduced lysozyme (10 mg/mL) was prepared by diluting appropriate volume of the enzyme in a denaturation buffer containing 6 M GdmCl and 40 mM DTT in 50 mM potassium phosphate buffer, pH 7.1. The sample was incubated at RT for a period of 12 hours to yield a fully reduced denatured enzyme. The denatured reduced lysozyme was used for refolding studies.

**Aggregation accompanying refolding.** Refolding of lysozyme was initiated after 50-fold rapid dilution of the denatured enzyme solution (10 mg/mL) with a refolding buffer (5 mM GSH, 5 mM GSSG in 50 mM potassium phosphate buffer, pH 8.5) containing artemin at the final concentrations of 0.5 and 1 μg/mL. The concentrations of the redox reagents were in the range of the optimum concentrations suggested by a previous report [21]. The final concentration of lysozyme was 13.89 μM in the buffer. The kinetic of refolding was recorded by monitoring light scattering at 400 nm at RT for 2 min using a microplate spectrophotometer (μQuant, BioTek, USA). The denatured lysozyme refolded without artemin was used as control. In these experiments, H-artemin (25–80˚C), O-artemin (0–100 mM $H_2O_2$), and HO-artemin (25–100 mM $H_2O_2$, 25–70˚C) were used.

## Results

### Temperature-dependent structural changes in artemin

Intrinsic fluorescence was monitored at 330 nm with 280 nm excitation to probe changes in tertiary structures of H-artemin (Fig 1A). The results indicated that when the temperature increased up to 30˚C, the fluorescence intensity decreased sharply, and the lowest emission intensity was recorded for the heated protein at 80˚C (Fig 1A). Besides, the wavelength of maximum emission ($\lambda_{max}$) did not show any shift.

The fluorescence of 30 μM ANS in the presence of H-artemin was also monitored at 470 nm after excitation at 380 nm (Fig 1B). ANS binding showed that the fluorescence intensity did not change considerably from 25 to 60˚. In contrast, a sharp emission peak was monitored at 470 nm for the heated proteins incubated at temperatures 70–80˚C.

Glutaraldehyde cross-linking of artemin followed by SDS-PAGE was performed for the heated proteins (Fig 2A). Analysis of the areas of dimeric bands in each experiment using

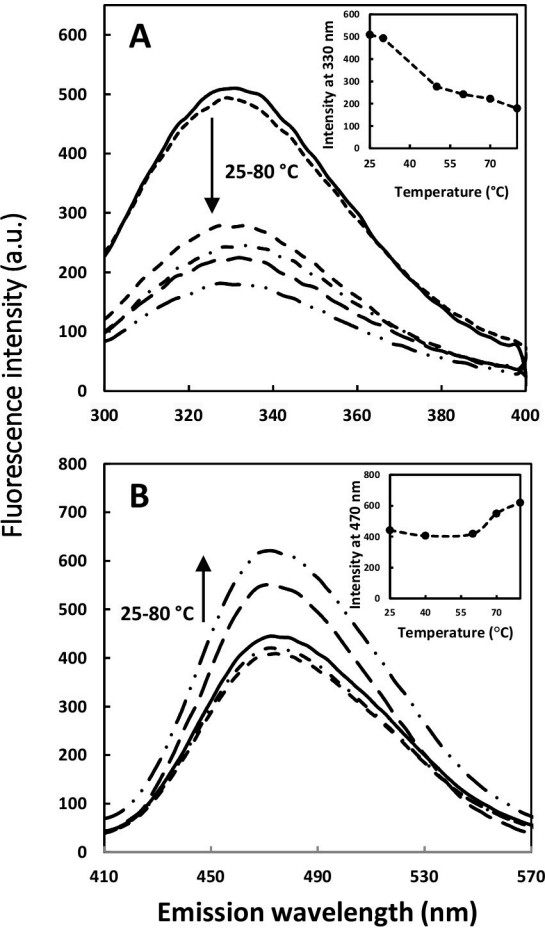

**Fig 1.** Intrinsic (A) and extrinsic (B) fluorescence emission intensity of artemin at different temperatures. A) The fluorescence emission of artemin was measured by incubating artemin (3.7 μM in 50 mM phosphate buffer, pH 7.2) at 25–80°C for 20 min, followed by cooling down at RT for 20 min. The fluorescence intensity was recorded with the excitation wavelength at 280 nm. The inset is temperature dependence of the fluorescence intensity of artemin at 330 nm. B) Fluorescence emission intensity of ANS in the presence of heated artemin. Samples of artemin (9.26 μM in 50 mM phosphate buffer, pH 7.2) were incubated at 25–80°C for 20 min, and cooled down at RT, then ANS (30 μM at final concentration) was added and fluorescence intensity was measured with the excitation wavelength at 380 nm. The inset indicates temperature dependence of the fluorescence intensity of artemin at 470 nm.

ImageJ software revealed that upon increasing temperatures from 25 to 50°C, the band corresponding to dimeric form of artemin around 54 kDa slightly strengthened and after that, it was reduced at 60°C, and finally completely disappeared at 70 and 80°C (Fig 2B). Besides, the dimeric band disappeared when artemin was incubated with DTT before heat treatments under both cross-linking and non-cross-linking conditions showing that induced dimerization of H-artemin is dependent on the formation of disulfide bond(s) (Fig 2C and 2D).

DLS measurement confirmed that the average size diameter of the heated chaperone increased upon enhancement of the temperature (Fig 3) and accordingly, the largest size was detected at 70 and 80°C. The larger size distribution of H-artemin at elevated temperatures can be explained by its ability to form oligomeric species as well as large aggregates as also indicated by SDS-PAGE (Fig 2A). All results showed that the conformational changes occur in artemin upon increasing temperature.

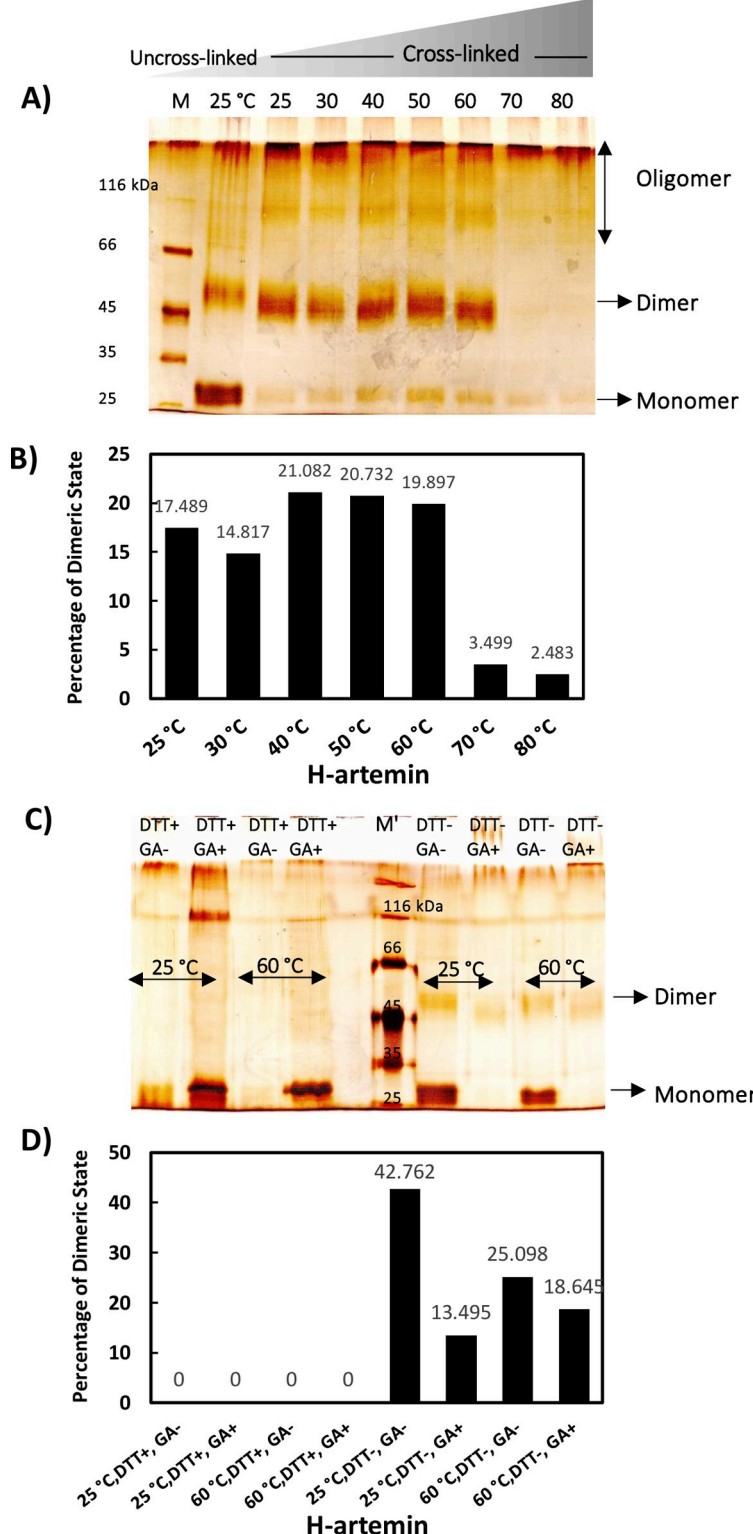

**Fig 2. Protein cross-link analysis of heated artemin by non-reducing SDS-PAGE.** A) Glutaraldehyde cross-linking of H-artemin and B) the levels of the 54 kDa bands corresponding to the dimers determined using ImageJ software. Artemin (7.41 μM) was incubated at 25–80°C for 20 min, then cooled down at RT for 20 min. Glutaraldehyde (0.5%) was added and the reactions were terminated after 1 min by addition of 200 mM Tris-HCl, pH 8.0. The uncrossed artemin incubated at 25°C was used as control. C) The heated protein fractions treated with(out) DTT as the reducing

agent. 30 mM DTT was added to artemin (3.7 μM), followed by incubation of the samples at 25 and 60°C for 20 min. The (un)cross-linked (GA$^{(-)+}$) (non)reduced (DTT$^{(-)+}$) artemin was analyzed using 10% non-reducing SDS-PAGE. M; protein standard marker. D) The levels of the dimeric bands were densitometrically determined using ImageJ software.

## Oxidative-dependent structural changes in artemin

As depicted in Fig 4A, fluorescence emission maximum ($\lambda_{max}$), as well as the fluorescence intensity of protein samples, were influenced by increasing the oxidant concentration. The slight decrease in fluorescence at 332 nm was monitored with increasing the concentration of oxidant from 2.5 to 40 mM, followed by a considerable decline in fluorescence intensity for O-artemin with 80–160 mM $H_2O_2$. Besides, $\lambda_{max}$ shifted to longer wavelengths, from 330 to 337 nm (Fig 4A).

Extrinsic fluorescence using ANS showed a two-state process. The O-artemin with lower concentrations of $H_2O_2$ (0–40 mM), showed slight intensity decreases, but the intensity was enhanced at higher concentrations of the oxidant (80–160 mM). As depicted in Fig 4B, a 10 nm red shift of the emission peak position of O-artemin was monitored when spectra at lower oxidant contents (0–40 mM, ~ 474 nm) are compared to those at higher oxidant concentrations (80–160 mM, ~ 484 nm).

SDS-PAGE and ImageJ analysis of cross-linked artemin showed that the band corresponding to dimeric form of O-artemin gradually strengthened upon increasing the concentration of $H_2O_2$ from 0 to 40 mM, and it was weakened at 100 and 160 mM $H_2O_2$ (Fig 5A and 5B).

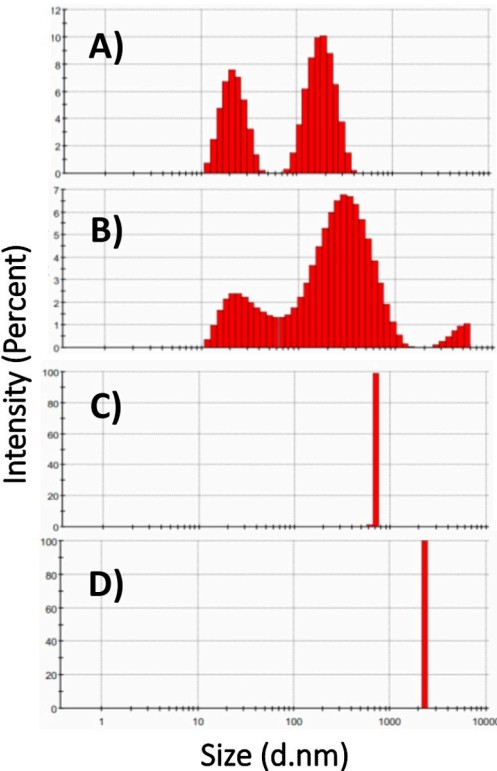

**Fig 3. DLS analysis of heat-treated artemin.** Artemin was incubated at 25 (A), 50 (B), 70 (C), 80 (D)°C for 20 min, then kept at RT for 20 min and size distribution analysis of H-artemin was performed.

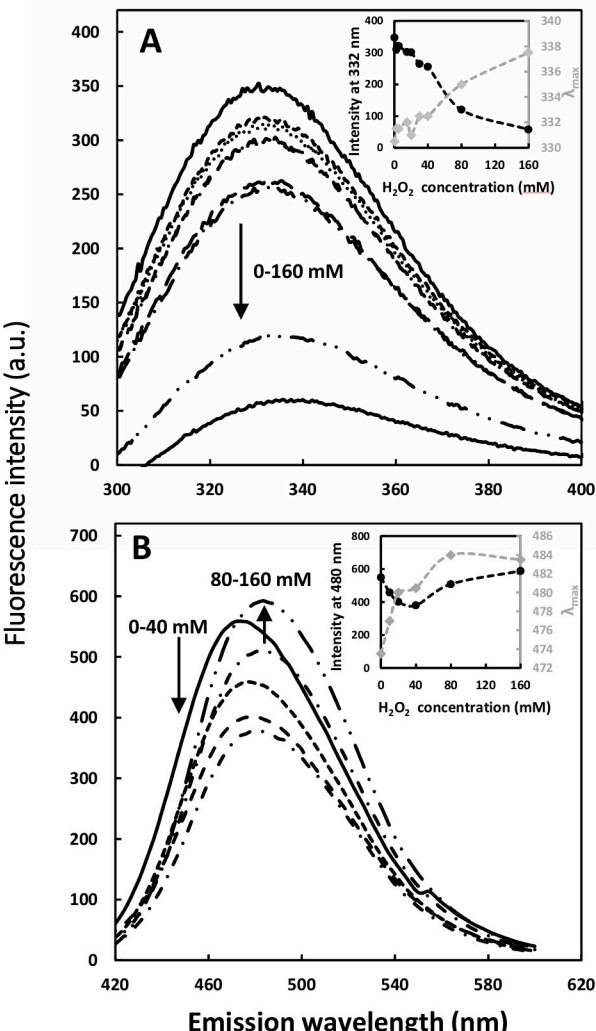

**Fig 4.** Intrinsic (A) and extrinsic (B) fluorescence emission intensity of artemin at different $H_2O_2$ concentrations. A) Intrinsic fluorescence emission of artemin was measured by incubating artemin (3.7 μM in 50 mM phosphate buffer, pH 7.2) at 0–160 mM $H_2O_2$ for 6 h at 0°C in dark, then the fluorescence intensity was recorded with the excitation wavelength at 280 nm. The inset is the oxidant dependence of the fluorescence intensity of artemin at 330 nm. B) ANS fluorescence emission spectra was monitored after addition of 30 μM ANS to O-artemin (9.26 μM in 50 mM phosphate buffer, pH 7.2). The excitation wavelength was 380 nm. The inset shows the oxidant dependence of the fluorescence intensity of artemin at 480 nm and also shifting the maximum emission wavelength ($\lambda_{max}$) of the oxidized chaperone from 474 to 484 nm.

### Structural changes of artemin under heat and oxidative conditions

To evaluate the influence of the two stressors, i.e. heat and oxidant, on protein structure, artemin was treated with 0–100 mM $H_2O_2$, followed by exposure to 50 and 70°C. Intrinsic fluorescence measurements showed that under both temperature incubations, the intensity declined gradually by increasing the oxidant concentrations from 0 to 100 mM (Fig 6A). Besides, ANS fluorescence indicated that the fluorescence intensity did not change considerably for HO-artemin incubated with 10–100 mM $H_2O_2$ at elevated temperatures (Fig 6B). This trend was not similar to those observed for the individual oxidant treatments (Fig 4B).

In a second approach, changes in the oligomerization state of HO-artemin were checked by SDS-PAGE (Fig 7A). Analysis of the areas of dimeric bands in each experiment by ImageJ

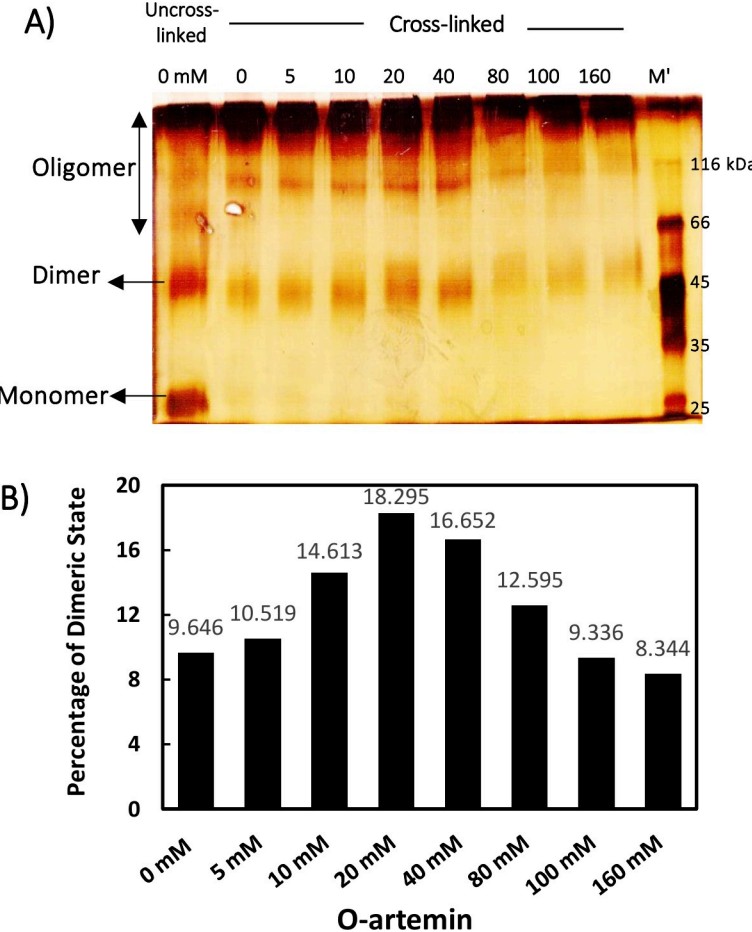

**Fig 5.** A) Protein cross-link analysis of oxidized artemin with various $H_2O_2$ concentrations by SDS-PAGE and B) the levels of the dimeric bands measured by ImageJ software. Purified artemin (7.41 μM) was incubated with 0–160 mM $H_2O_2$ for 6 h at 0°C in dark conditions. Before the samples were loaded on 10% non-reducing SDS-PAGE, a trace amount of catalase (8.33 nM) was added to the protein solutions to decompose the remaining $H_2O_2$. M; protein standard marker.

software showed that the bands representing the dimeric forms of HO-artemin were clearly strengthened along with enhancement of the oxidant concentrations (Fig 7B).

## Structural changes of artemin under stress conditions are irreversible

To determine whether the temperature/oxidation-dependent transition of artemin is reversible, artemin was incubated at elevated temperatures (H-artemin), high oxidant concentrations (O-artemin), and both heat and oxidant conditions (HO-artemin), followed by elimination of the heat and oxidant and keeping the samples at 4°C for 48 h before fluorescence measurements (Fig 8). The results showed that fluorescence intensity did not change after 48 h. In fact, the structure of artemin modified irreversibly upon exposing to the stressors.

## Chaperone-like activity of artemin under heat and oxidative stress

**H-artemin.** Denatured/reduced lysozyme was refolded by the dilution method and the kinetics of chaperone-assisted refolding was examined in the presence of 0.5 and 1 μg/mL artemin (Fig 9). As shown in Fig 9A and 9B, the heated artemin at 25 and 50°C was found efficient

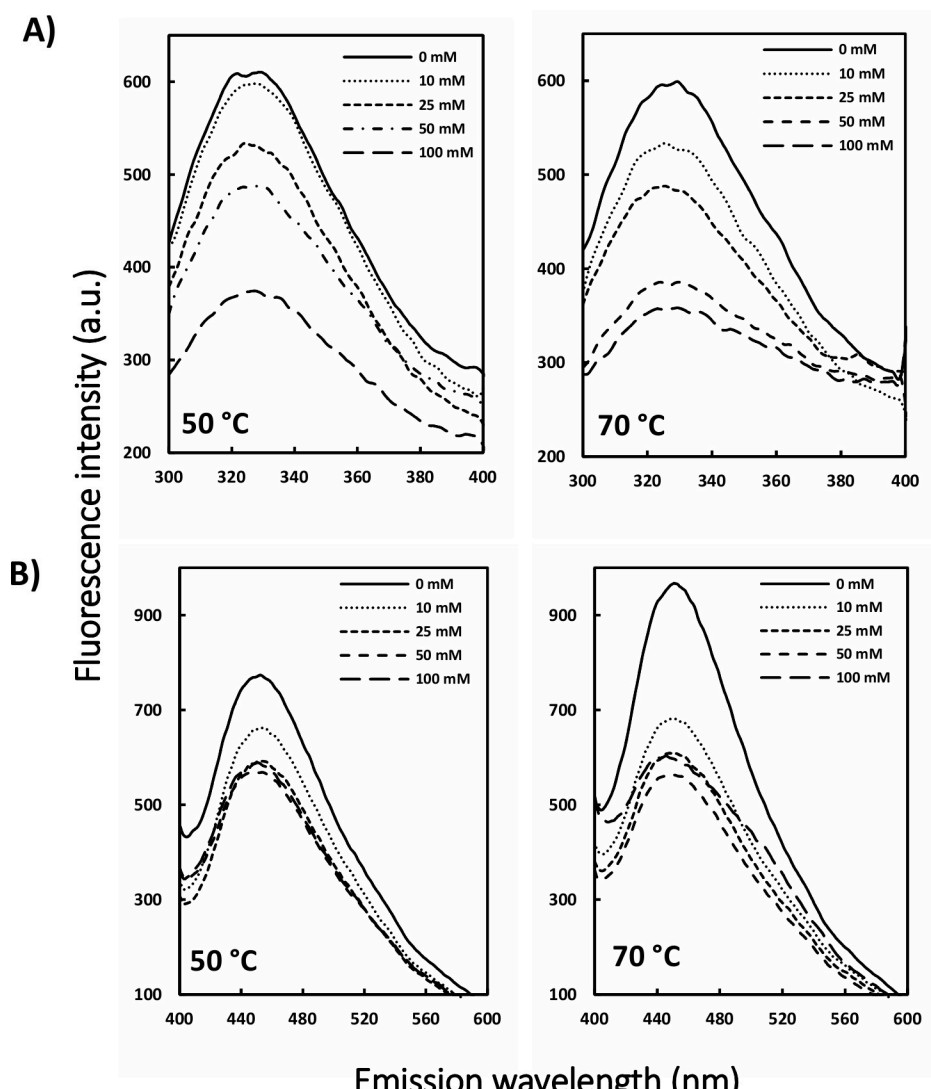

**Fig 6.** Intrinsic (A) and extrinsic (B) fluorescence emission intensity of heated oxidized artemin. A) Intrinsic fluorescence emission of artemin was recorded by incubating artemin (3.7 μM in 50 mM phosphate buffer, pH 7.2) with 0–100 mM $H_2O_2$ for 6 h at 0˚C in dark, followed by incubating the samples at 50 and 70˚C for 20 min, then cooling down at RT. The fluorescence intensity was recorded with the excitation wavelength at 280 nm. B) Fluorescence emission intensity of ANS (30 μM at final concentration) was measured in the presence of HO-artemin (9.26 μM) with the excitation wavelength at 380 nm.

in suppressing aggregation of the enzyme. In contrast, 0.5 μg/mL chaperone incubated at elevated temperatures, 60 to 80˚C, accelerated the aggregation of lysozyme (Fig 9A), and 1 μg/mL artemin showed no effect on the refolding yield at similar conditions (Fig 9B).

**O-artemin.** Results showed that the oxidized chaperone (0.5 μg/mL) with 25 and 50 mM $H_2O_2$ could partly suppress aggregation of lysozyme compared to the untreated control (Fig 9C). In contrast, the anti-aggregatory potency of O-artemin weakened by increasing the oxidant content, and the oxidized chaperone with 100 mM $H_2O_2$ did not show any effect on the enzyme refolding yield (Fig 9B and 9C).

**HO-artemin.** Fig 10 depicts the impact of HO-artemin (1 μg/mL) on refolding of lysozyme based on constant concentrations of $H_2O_2$ (25, 50 and 100 mM) (Fig 10A, 10B and 10C)

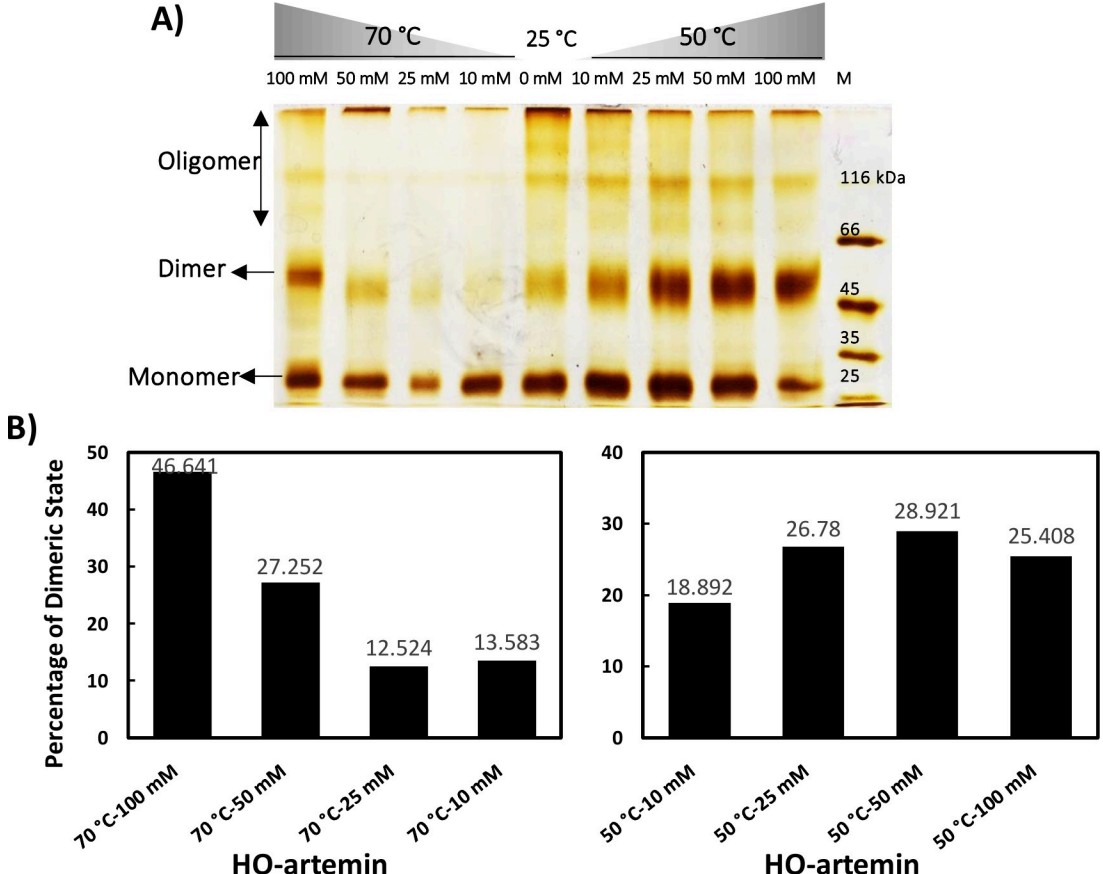

**Fig 7.** A) Cross-link analysis of heated oxidized artemin by SDS-PAGE and B) the levels of the dimeric bands measured by ImageJ software. Purified artemin (7.4 μM) was incubated with 0–100 mM $H_2O_2$ for 6 h at 0˚C in dark condition. Then the samples were subjected to temperatures of 50 and 70˚C for 20 min and cooled down at RT. Before the samples were loaded, catalase (8.33 nM) was added to the solutions to decompose the remaining $H_2O_2$.

and temperatures (25, 50 and 70˚C) (Fig 10D, 10E and 10F). According to Fig 10A, the HO-artemin at 25 and 50˚C showed a potency in suppressing the aggregation of lysozyme, but at the elevated temperature, 70˚C, it could not efficiently prevent the aggregation of lysozyme (Fig 10A and 10B). These trends were similar to the graphs obtained for H-artemin (Fig 9A and 9B). In contrast, HO-artemin incubated with 100 mM $H_2O_2$ clearly exhibited an improved chaperone activity (Fig 10C).

We also checked the effect of HO-artemin on refolding of lysozyme based on constant temperatures (Fig 10D, 10E and 10F). Results clearly demonstrated an enhanced capacity of the chaperone along with the increased oxidant concentrations at 25˚C (Fig 10A). At higher temperatures (50 and 70˚C), oxidized artemin with 50 and 100 mM $H_2O_2$ showed a similar rate (Fig 10B and 10C). Comparing these observations with results from the effect of O-artemin on refolding yield (Fig 9C and 9D) reveals that the chaperoning function of HO-artemin was considerably improved in the presence of both stressors.

## Discussion

Artemin is an abundant heat stable protein in *Artemia* encysted embryos and it was found that high regulatory production of artemin under harsh environmental conditions is probably

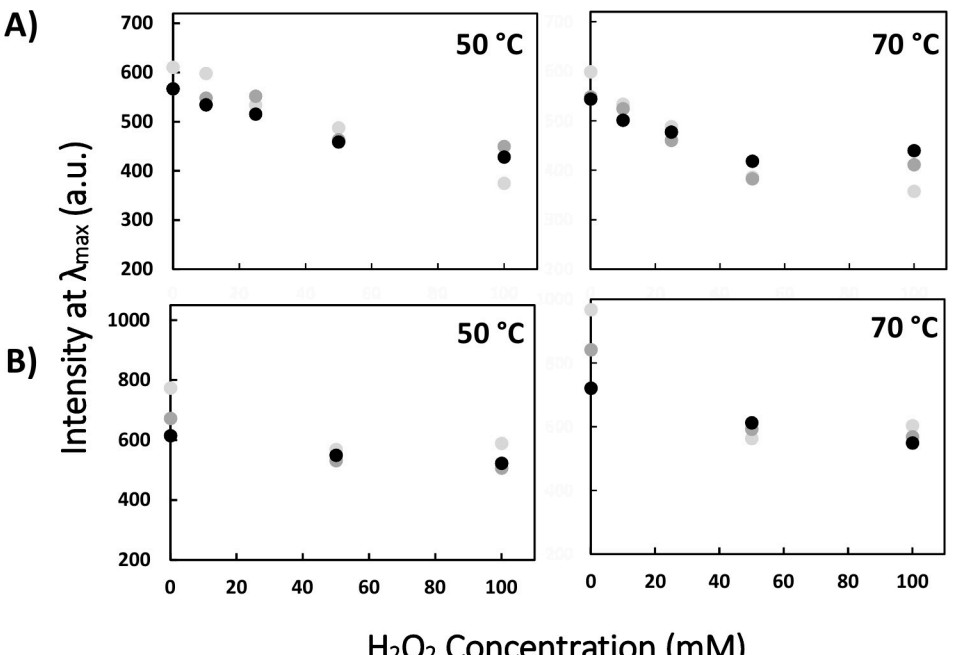

**Fig 8. Structural changes of artemin under different stress conditions are irreversible.** A) Artemin at final concentration of 3.7 μM was treated with 0–100 mM $H_2O_2$ for 6 h at 0°C in dark, followed by incubating at 50 and 70°C for 20 min. Then the samples were kept at 12°C for 48 hours and the fluorescence intensity was recorded with the excitation wavelength at 280 nm and the fluorescence intensity at $\lambda_{max}$ was determined. B) ANS (30 μM) fluorescence intensity at $\lambda_{max}$ in the presence of 9.26 μM HO-artemin with the excitation wavelength at 380 nm.

relevant to stress resistance in this crustacean [4, 22]. Artemin is not only capable of suppressing heat-induced aggregation of different protein substrates such as citrate synthase, carbonic anhydrase, horseradish peroxidase and luciferase *in vitro* and *in vivo*, but also protecting nucleic acids as a potent chaperone [8–11, 13]. The intrinsic conformational properties of artemin seem to play crucial roles in its biological activities. Artemin showed a higher surface hydrophobicity when compared to other molecular chaperones, based on our previous studies [11]. Moreover, the chaperoning potency of artemin is greatly dependent on the presence of cysteine residues and intermolecular disulfide bond formation [14]. Accordingly, heat and oxidation are among the most important factors, which probably affect the chaperone capacity of artemin. In this report, the conformational transitions of artemin have been evaluated under heat and oxidative stress using different structural and functional analysis. The results have been summarized in Table 1. Our findings may shed light on the mechanisms by which small heat shock proteins (sHsps) could play their protective roles in various stress conditions. Notably, the structural changes of artemin under oxidative stress conditions have not been investigated to date.

Different mechanisms have been suggested for sHsps to protect protein substrates in different stresses [23]. Many sHsps are known to undergo temperature-dependent structural alterations, which modulate their chaperone activities [24]. These mechanisms may be mediated through conformational transitions of the dimeric and oligomeric forms of the chaperones to each other. For example, at elevated temperatures, dissociation of large oligomeric forms of sHsp26 into smaller, active species of the chaperone occurs [25]. Also temperature-dependent rearrangements in the tertiary structure of Hsp22 resulted in its improved chaperone activity [26]. The chaperoning potency of a redox-regulated chaperone, Hsp33, was mainly mediated

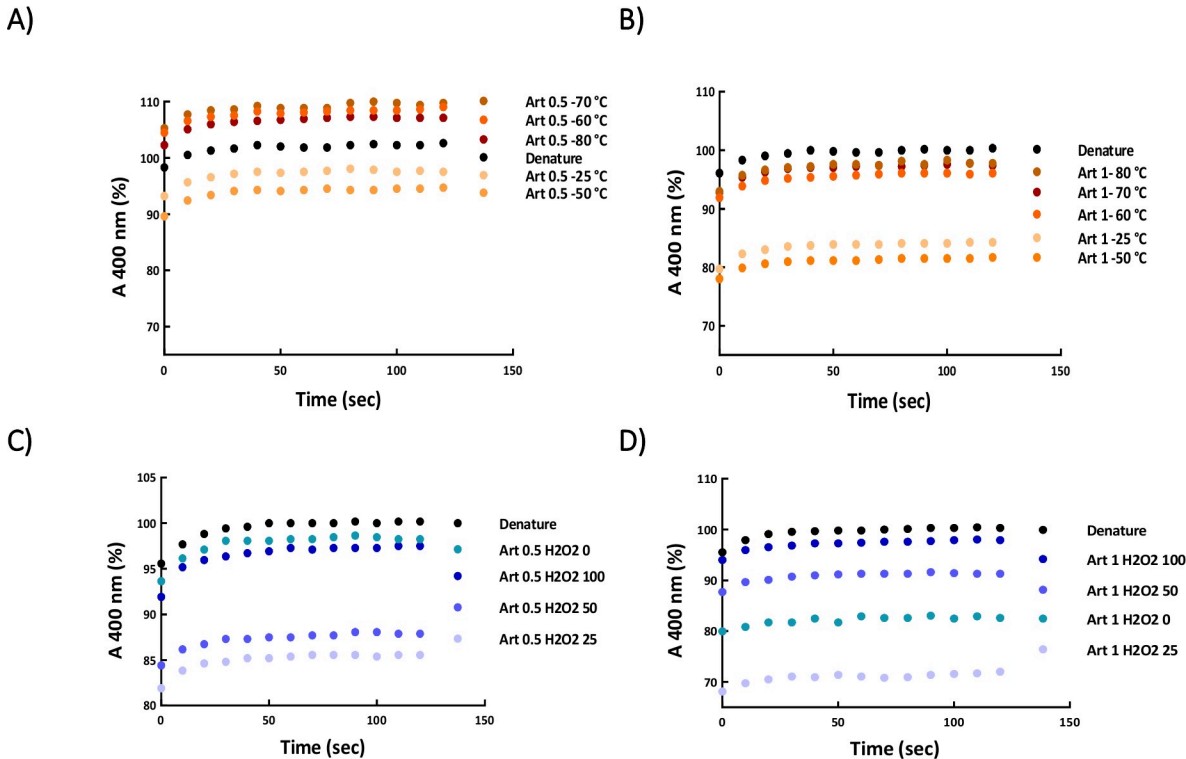

**Fig 9.** Refolding of lysozyme in the absence (●) and presence of (●) heated (A, B) and (●) oxidized (C, D) artemin. Denatured lysozyme (10 mg/mL) in a solution containing 40 mM DTT, 6 M GdmCl, and 50 mM potassium phosphate buffer, pH 7.1, was diluted with a mixing ratio of 1:50 by the refolding buffer. The diluted sample contained 13.89 μM lysozyme, 0.8 mM DTT, 0.12 mM GdmCl, and 50 mM potassium phosphate buffer, pH 8.5, 5 mM GSH, 5 mM GSSG, and 0.5 (A, C) and 1 μg/mL (B, D) artemin. The kinetic of refolding was recorded by monitoring light scattering at 400 nm at 25˚C.

by the heat effect and its monomer-dimer switch [27]. Besides, Hsp20.1 and Hsp14.1 oligomers dissociated to smaller oligomeric forms or even dimer/monomer species under acid stress [23]. The dissociation seems to be necessary for the exposure of additional hydrophobic sites on the surface of the protein molecule [28].

## Heat-dependent structural transitions

At elevated temperatures, artemin has been expected to undergo heat-dependent structural transitions like other sHsps, leading to exposure of a large amount of hydrophobic residues on the protein surface. Since artemin contains seven Trp and five Tyr residues (S1 Fig), fluorescence studies could be applied to monitor its conformational changes. At present, there is no reliable information for the accurate spatial localization of these amino acids in artemin. It has been shown that Trp and Tyr fluorescence is usually quenched in a distance-dependent manner. Accordingly, when such aromatic amino acids are located on the surface of the protein, the fluorescence intensity of the residues is usually influenced in a higher degree by the quencher, compared to the amino acids deeply embedded in the protein structure [29]. As depicted in Fig 1A, the fluorescence emission of the protein was significantly reduced at elevated temperatures and the first transition was observed at 50˚C, followed by the second transition at 80˚C. It can be suggested that the induced conformational changes of artemin resulted in the increased quenching, probably due to the exposure of the buried aromatic residues to the solvent.

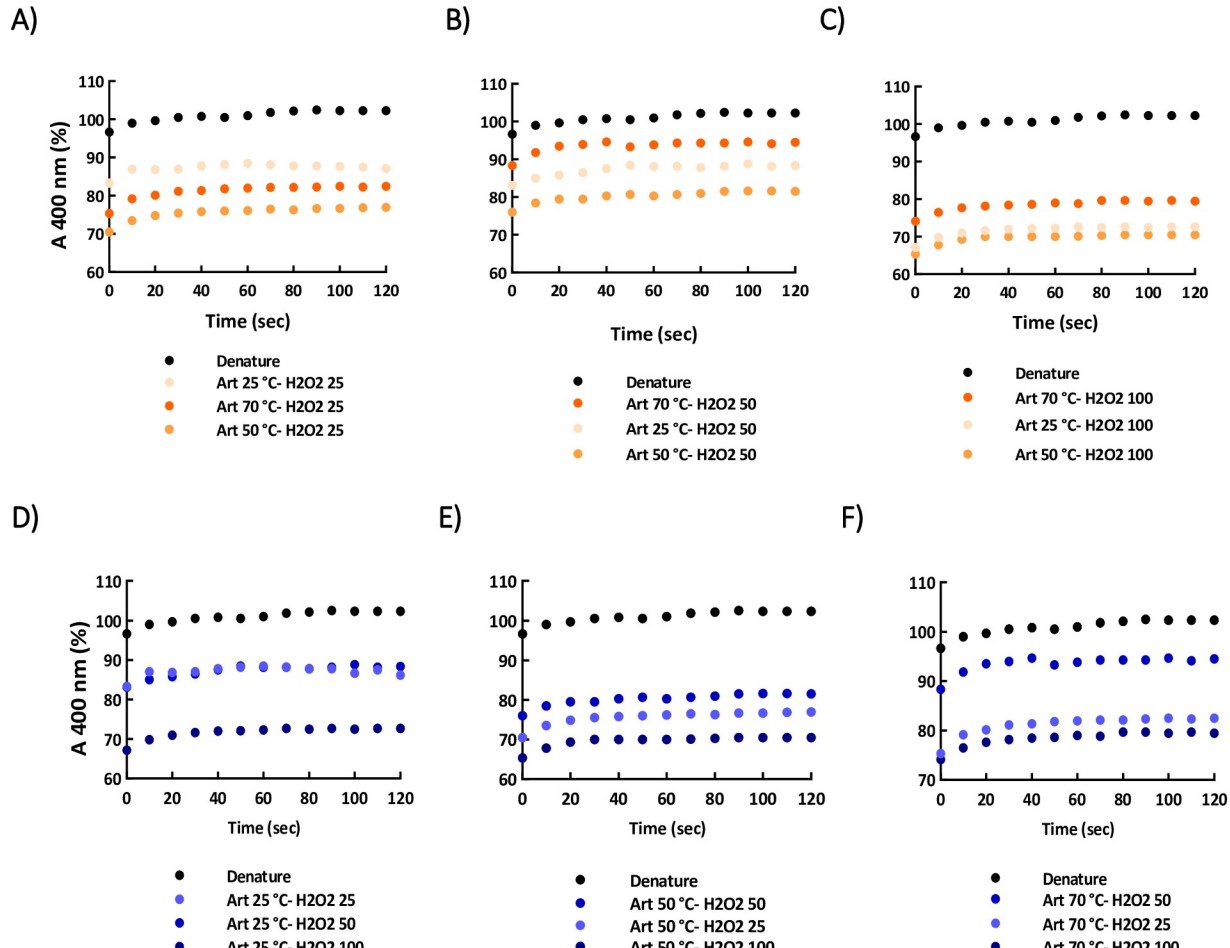

**Fig 10. Effects of heated-oxidized artemin on aggregation accompanying refolding of lysozyme.** The comparison was performed based on constant (●) $H_2O_2$ concentrations (A-C) and (●) temperatures (D-F). Artemin at a final concentration of 1 μg/mL was used and the denatured sample without added chaperone was introduced as control (●). The kinetic of refolding was recorded by monitoring light scattering at 400 nm at 25°C.

The conformational changes of the protein was also investigated using ANS fluorescence studies. ANS is widely used as a hydrophobic fluorescent probe for monitoring hydrophobic patches of proteins [28, 30]. It is a non-fluorescent probe in aqueous solutions, while it becomes highly fluorescent in non-polar environment [28]. Increasing temperature up to

**Table 1. A summarize of the structural and functional changes of artemin upon heat and $H_2O_2$ treatments.**

| Stress / Changes | Heat | $H_2O_2$ | Heat & $H_2O_2$ |
|---|---|---|---|
| **Hydrophobicity** | 70–80°C: ↑ | 0–40 mM: ↓ | 50, 70°C; 10–100 mM: ↓ |
| | | 80–160 mM: ↑ | |
| **Dimerization** | 25–50°C: ↑ | 0–40 mM: ↑ | 50, 70°C; 10–100 mM: ↑ |
| | 60–80°C: ↓ | 80–160 mM: ↓ | |
| **Reversibility** | Irreversible | Irreversible | Irreversible |
| **Chaperone Activity** | 25–50°C: Less to More | 0–25 mM: Less to More | 50–70°C; 25–100 mM: |
| | 60–80°C: More to Less | 50–100 mM: More to Less | Less to More |

60˚C lead to higher exposure of artemin non-polar regions and binding ANS to these hydrophobic sites, where an enhanced fluorescence intensity and a weak blue shift in the $\lambda_{max}$ (Fig 1B) were observed, similar to other reports [28, 31]. Such hydrophobic interactions in artemin likely play a critical role in acquiring its functional conformation and mediating the protein-chaperone interactions [30]. In contrast, no change was observed in fluorescence intensity, when the protein was incubated at lower temperatures, presumably reflecting negligible changes in the protein surface hydrophobicity, particularly the lack of ANS-binding sites. Due to the thermal inactivation of the excited state of the probe, the heat-treated proteins allowed to cool down at RT for 20 min followed by fluorescence measurements [30].

In a second approach, different monomer, dimer and oligomer forms of H-artemin were analyzed under thermal conditions using SDS-PAGE technique after addition of 0.5% glutaraldehyde as a fixative agent. Glutaraldehyde cross-linking is a commonly used method for determination of the subunit structure of oligomeric proteins [32]. The areas of the dimeric bands of the heated protein on SDS-PAGE were further checked using ImageJ software. SDS-PAGE and ImageJ analysis showed that the uncross-linked protein was mainly presented as monomers at 25˚C (Fig 2A and 2B), while the protein was mostly observed in dimeric forms under cross-linking condition. The abundance of the 54 kDa protein band was slightly enhanced upon increasing temperatures from 25 to 50˚C, whereas it became significantly weaker at 60˚C and completely disappeared at 70 and 80˚C, probably due to the formation of higher molecular weight oligomers and/ or aggregates *via* hydrophobic interactions that could not enter the gel (Fig 2A). Furthermore, we examined whether the heat-induced dimerization is mediated *via* disulfide bonds by addition of DTT as a reducing agent to the protein solutions before the heat treatment. Accordingly, the dimers were disappeared in the presence of DTT, confirming that the disulfide bond(s) are necessary for dimerization (Fig 2C and 2D). Size distribution analysis of H-artemin showed that the size of proteins was enhanced upon increasing temperatures, probably due to the formation of dimers, oligomers (Fig 3A and 3B) and protein oligomeric assembly and/or aggregates (Fig 3C and 3D) resulted from intermolecular hydrophobic interactions (Fig 1B).

Many previous reports revealed that N-terminus of sHsps is rich in hydrophobic residues and involved in binding client proteins due to the presence of these hydrophobic residues in this region, which is located on the inner surface of the oligomers [33]. Therefore, heat-induced oligomer dissociation may lead to the exposure of a large number of hydrophobic surfaces that are normally buried in the oligomer, including not only the N-terminus, but also the dimer–dimer and even the monomer–monomer interfaces [23]. As a result, the substrate-binding ability of the chaperones would be improved in a temperature-dependent manner to cope with the increasing number of the client substrates within cells [34]. Totally, our results revealed that at elevated temperatures, H-artemin presumably undergoes the structural changes, which are associated with a marked increase in the surface hydrophobicity and also degree of dimerization of the chaperone *via* disulfide bridges in order to improve the binding capacity of the chaperone to the client proteins in an unfolding intermediate state. As a suggestion, interaction of the protein substrates may subsequently prevent the self-association of the molecular chaperone with a highly exposed hydrophobic surface during the stress conditions.

## Oxidative-dependent structural transitions

Since artemin contains a high content of cysteine residues with specific distributions, cysteines may be regarded as critical factors in regulating chaperone function [3, 15]. We have shown previously that oxidation/reduction of cysteine residues greatly influences chaperone potency of artemin [14]. Experimental results revealed that 9 out of 10 thiols are free in artemin

monomers and there is one cysteine involved in inter-molecular disulfide bond formation. Moreover, our molecular modeling studies predicted that Cys22, with the highest accessible surface area, is likely the responsible residue for inter-subunit disulfide bond formation [14]. Accordingly, it was supposed that the formation of an inter-molecular disulfide bond between two monomers of artemin leads to its dimerization and switching between its less and more active forms. Here, we tried to provide detailed structural information on conformational changes of artemin during oxidative conditions. Intrinsic fluorescence measurements demonstrated that upon increasing hydrogen peroxide concentrations to 40 mM, artemin structure was still stable (Figs 4A and 5), when compared with many other oxidant susceptible proteins [35, 36]. However, higher concentrations of the oxidant (80–160 mM) led to a remarkable reduction in the fluorescence intensity of O-artemin with a slight red shift of $\lambda_{max}$ (Fig 4A, insert), probably due to the changing in the microenvironment around the aromatic residues during oxidation. Trp and Tyr residues show high tendency to interact with negatively charged residues, cysteine and disulfide bonds in their spatial environments, where the interactions of sulfur atoms with their aromatic rings may lead to the fluorescence decay [37]. Cystines that are involved in disulfide bonds have a high propensity to interact with tryptophan residues [38]. Due to the micro-environment alteration of the fluorescent residues resulted from the breaking or forming of the disulfide bonds, the fluorescence properties of proteins could be changed [39]. Accordingly, variable degrees of fluorescence quenching of Trp and Tyr residues probably occurred by nearby formed disulfide and sulfhydryl groups in O-artemin (Fig 4A). Moreover, the quenching of Trp fluorescence upon exposing to the oxidant may also occur as a result of oxidation of the side chains of the aromatic residues and forming a range of aromatic products in oxidized proteins [40].

ANS binding indicated that increasing $H_2O_2$ content from 10 to 40 mM led to a decline in the fluorescence intensity (Fig 4B), as well as a shift in $\lambda_{max}$ to longer wavelengths (Fig 4B, insert). Therefore, it seems that the surface hydrophobicity of O-artemin has been relatively decreased. It is suggested that oxidation of Cys residues and the subsequent disulfide bond formation resulted in oligomerization of artemin, where the hydrophobic patches were simultaneously buried deeply within the protein oligomer assemblies. Although, higher concentrations of the oxidant (80–160 mM) resulted in an increase in ANS binding, the increase was not as significant as that observed for H-artemin (Fig 1B). Similar trends were also recorded for the chaperone GroEL in the presence of $H_2O_2$ [41].

Glutaraldehyde cross-linking of the oxidized artemin in the presence of 0–160 mM $H_2O_2$ was carried out to further confirm the dimerization/oligomerization status of O-artemin. SDS-PAGE and ImageJ analysis demonstrated that the 54 kD dimeric bands were gradually sharpened along with increasing the oxidant content from 0 to 40 mM (Fig 5), and the bands were weakened at 100 mM $H_2O_2$, probably due to the formation of a higher degree of oligomers (Fig 5A and 5B). It is hypothesized that the highly reactive thiol groups of cysteine residues may lead to protein oligomerization due to the excess formation of disulfide bonds in O-artemin. Finally, the protein precipitation may occur as a result of highly exposed hydrophobic regions and the formation of excess inter-protein disulfide bridges [38], as showed by fluorescence (Fig 4B) and SDS-PAGE analysis (Fig 5). Taken together, the results indicate that upon oxidation, artemin undergoes dimerization and oligomerization.

## Structural transitions of artemin under heat and oxidant treatments

Finally, to examine the simultaneous effect of stressors on protein structure, artemin was exposed to different concentrations of $H_2O_2$ (0–100 mM), followed by incubation at elevated temperatures (50, 70˚C). The fluorescence measurements showed that the intensity of

emission decreased upon increasing the oxidant content at both temperatures (Fig 6A). This result is in a good agreement with the fluorescence measurement of O-artemin (Fig 4A). Based on ANS binding studies, a decreasing trend for fluorescence intensity of HO-artemin was observed (Fig 6B) along with the increased oxidant concentrations (up to 100 mM $H_2O_2$). However, O-artemin showed an increased ANS binding upon treatment with 80–160 mM $H_2O_2$ (Fig 4B). Our suggestion is that the oxidant agent sequestered the exposed hydrophobic surfaces of the heated chaperone by stabilizing the dimeric forms, thereby preventing the protein from aggregation. Besides, we checked the tertiary structural changes of HO-artemin using SDS-PAGE analysis of the cross-linked protein (Fig 7A and 7B). In agreement with the results obtained for O-artemin (Fig 5) and H-artemin (Fig 2A), the dimerization of HO-artemin was enhanced upon increasing the oxidant concentrations from 0–100 mM $H_2O_2$ at both temperature treatments, and the degree of dimerization was higher at 50°C in comparison with 70°C (Fig 7).

The reversibility of the structural changes of artemin was investigated using fluorescence measurements (Fig 8). Both intrinsic (Fig 8A) and extrinsic (Fig 8B) fluorescence analyses confirmed the irreversible structural alteration of artemin under heat and oxidative conditions.

## Heat/oxidative-dependent chaperone activity

To assess the potency of heated/oxidized artemin on the aggregation accompanying refolding of lysozyme, the refolding experiments were performed in the presence of 0.5 and 1 μg/mL artemin. Our results indicated that H-artemin incubated at 25 and 50°C extensively promoted the refolding of the enzyme (Fig 9A and 9B), however, at elevated temperatures (60–80°C), it accelerated the enzyme aggregation (Fig 9A). In addition, oxidation with lower concentrations of $H_2O_2$ (25, 50 mM) made the chaperone more active and subsequently promoted the refolding of the enzyme, but, the higher degree of artemin oxidation of (in the presence of 100 mM $H_2O_2$) did not change the lysozyme refolding yield (Fig 9C and 9D). Our suggestion is that the higher concentrations of the oxidant influenced the structure of artemin and a weak chaperone activity was observed as a result of oxidative modification of some amino acids of the protein, and alteration of the protein surface hydrophobicity [41], leading to self-association/agglomeration of the chaperone. Fluorescence (Fig 4) and protein cross-link (Fig 5) analysis confirmed this suggestion.

Finally, the chaperone activity of HO-artemin was investigated. The results revealed that the ability of HO-artemin in preventing the enzyme aggregation was significantly improved (Fig 10). Analysis of data showed that the oxidant did not strongly influence the heated chaperone (Fig 10A and 10B). A similar anti-aggregatory trend was found for H-artemin and HO-artemin, especially at 25 and 50 mM $H_2O_2$ (Figs 9B and 10A–10C). In contrast, HO-artemin exhibited a higher efficacy in suppressing the enzyme aggregation (Fig 10D–10F) when compared to O-artemin (50 and 100 mM $H_2O_2$) (Fig 9D). Our observations revealed that although the presence of both heat and oxidant stressors is required to achieve the most active form of the chaperone, heat has a stronger impact on the structure and function of artemin than the oxidizing agent.

Here, we reported that the mechanism of activation of artemin mainly relies on protein dimerization, which is strongly modulated through heat-induced conformational changes of the protein. Besides, such dimer and/or oligomer structures can be stabilized through the formation of inter-subunit disulfide bridge(s) [42] in the dimer form and switch a less active to more active form of the protein. Subsequently, stable dimerization of artemin monomers leads to the accumulation of highly active artemin species. It has been indicated that disulfide bonds improve the thermal stability of proteins with an activity in the oxidizing extracellular

environment [38]. One explanation is that disulfide bonds reduce the conformational freedom and entropy of the protein in its unfolded state, and consequently destabilize this state with respect to the folded state [43]. Similar mechanisms have been proposed for other chaperones, Hsp33 and Hsp27, which control their dimerization by forming an intermolecular disulfide bond [27, 44]. Our observations suggested that there may be different mechanisms for artemin to protect client proteins in different stresses, which are basically mediated through protein's dimerization (Scheme 1). The proposed mechanisms presumably play a vital role in conserving the *Artemia* cysts's tolerance against severe environmental stresses.

**Scheme 1. Model of chaperone function of artemin.** Under physiological condition, artemin exists mainly in rosette-like oligomeric forms. Upon heat shock, the exposed hydrophobic patches of dimeric structures of the chaperone probably play a vital role in mediating the protein-chaperone interactions. At elevated temperatures, the highly exposed hydrophobic surfaces of the chaperone lead to its self-assembly as aggregates. The oxidized artemin also forms stable dimers through formation of disulfide bridges between the chaperone monomers and the protein oligomerization/agglomeration occurs as a consequent of the exposure of a high degree of hydrophobic surfaces and the formation of inter-protein disulfide bridges. Under both stress conditions, the most active stable form of the chaperone is achieved through formation of the stable dimers with an appropriate exposed hydrophobic sites. Through these mechanisms, artemin oligomers dissociate into dimers and monomers upon heat and/or oxidative stresses, which are able to bind non-native proteins, thus preventing their aggregation.

Our finding confirmed that artemin appears to exist in different conformational forms including monomer, dimer and oligomer as a function of heat and oxidant. Both hydrophobicity and dimerization are important to achieve the chaperone in a fully active form (S1 Graphical abstract). At elevated temperatures and higher degree of oxidation, both hydrophobicity and dimerization increased. In contrast, under both stress conditions, hydrophobicity did not change at the higher oxidant concentrations, while the dimerization was enhanced. Such dimerization strongly relied on disulfide bond formations between artemin monomers. The protein cross-link experiments indicated the oligomerization of artemin as a consequence of self-association at elevated temperatures. Our suggestion is that upon increasing temperature up to 60˚C, artemin exposes the maximum hydrophobic sites in order to stably bind the target folding intermediates. Therefore, the presence of such binding substrates may stabilize artemin conformation at higher temperatures and protect it from self-association/precipitation and this may result in the reversible structural changes of artemin upon exposing to stress conditions. Enhanced peptide-substrate binding upon heat treatment was previously reported for other molecular chaperones Hsp26 and gp96 [25, 45]. In the case of gp96, it was recognized that the heat-induced oligomers retain peptide binding ability and it was suggested that these soluble aggregates could serve as a reservoir, and be converted into activated chaperone molecules under certain circumstances [46]. Our aggregation accompanying refolding experiments also showed the chaperone potency was considerably influenced by the temperature rather than the oxidant. Also, the function of artemin was improved at the lower oxidant concentrations probably due to proper dimerization of the chaperone through disulfide bond formation as it was detected by cross-link analysis. Despite the weak chaperone potency of oxidized artemin with high concentrations of $H_2O_2$, simultaneous incubation of artemin at elevated temperatures with higher oxidant concentrations significantly triggered the activation of chaperone. As the fluorescence studies showed, the presence of the oxidant resulted in a lower ANS binding through sequestering the exposed hydrophobic patches on the heated artemin probably to inhibit its aggregation. It is suggested that oxidation presumably acts by stabilizing the dimer structures of artemin through formation of disulfide bridges between the protein monomers and strengthens its stability and chaperoning potency.

Urmia Lake is one of the most hypersaline lakes and the largest natural *Artemia* habitats in the world located in the northwestern region of Iran [47–49]. Lake water salinity used to fluctuate between 140 and 220 g/L before 1999, but the salinity of lake increased to 340 g/L during recent years [2] possibly due to drought and increased in agricultural water consumption [50]. The lake itself is also exposed to harsh and continental climate, with winter temperature reaching -20˚C and summer temperature of up to 40˚C [50]. One of the other adaptations, is the ability of *Artemia* cysts to tolerate anoxia for periods of years, while fully hydrated and at physiological temperatures [51]. Accordingly, salinity, heat and anoxia are the serious challenges that *Artemia* cysts face during their lifetime. Large quantities of artemin, which accumulate during encystation of *Artemia* embryos, and not degraded in cysts during such extreme stresses can partly justify the tolerance of *Artemia* cysts [52]. High salinity can induce the formation of reactive oxygen species (ROS) within cells, and its over accumulation results in oxidative damage of membrane lipids, proteins and nucleic acids [53]. Artemin is a redox regulated chaperone and it probably acts as a reducing reservoir in the cytoplasm of the brine shrimp *Artemia* to protect cells from oxidative damage. In addition, activation of artemin under stress conditions can protect *Artemia* embryos against heat, salinity and oxidative stress conditions.

## Conclusion

Our data suggest that the heat-induced dimerization of artemin is the most critical factor for its activation. It seems that *in vivo* cytosolic artemin may exist in a monomer–oligomer equilibrium in the cytoplasm of the cysts, which play a critical role in maintaining the cysts under such extreme conditions. Environmental stresses and/or intracellular portion of proteins may shift the equilibrium towards the active dimer forms. Moreover, the stabilization of artemin by disulfide bonds, has the potential to increase stress resistance, by protecting both RNA and protein substrates, in oviparously developing *Artemia* embryos. Further studies can be also performed in future for illustrating the substrate-chaperone interactions in stress conditions.

## Supporting information

**S1 Fig. Amino acid sequence of artemin from *A. urmiana* (GenBank accession no: EU380315.1).**
(DOCX)

**S1 File.**
(PDF)

**S1 Graphical abstract.**
(TIF)

**S1 Raw images.**
(PDF)

**S1 Scheme.**
(TIF)

## Acknowledgments

The authors appreciate Tarbiat Modares University for the instrumental and technical supporting the work.

## Author Contributions

**Conceptualization:** Reza H. Sajedi.

**Formal analysis:** Zeinab Takalloo, S. Shirin Shahangian.

**Investigation:** Zeinab Takalloo, Zahra Afshar Ardakani.

**Methodology:** Zeinab Takalloo, Zahra Afshar Ardakani, Bahman Maroufi.

**Project administration:** Zeinab Takalloo, S. Shirin Shahangian, Reza H. Sajedi.

**Supervision:** Reza H. Sajedi.

**Validation:** Zeinab Takalloo, Reza H. Sajedi.

**Writing – original draft:** Zeinab Takalloo.

**Writing – review & editing:** Zeinab Takalloo, Reza H. Sajedi.

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
