## [Decision Letter · Decision Letter 0]

18 Jun 2020

PONE-D-20-06479

Stress-Dependent Conformational Changes of Artemin: Eﬀects of Heat and Oxidant

PLOS ONE

Dear Dr. H. Sajedi,

Thank you for submitting your manuscript to PLOS ONE. After careful consideration, we feel that it has merit but does not fully meet PLOS ONE’s publication criteria as it currently stands. Therefore, we invite you to submit a revised version of the manuscript that addresses the points raised during the review process.

After careful evaluation of your manuscript and based on the comments received from the potential reviewers, I decided to give your manuscript a further chance.  Please consider all the comments/suggestions, especially those of the reference 1.  Also, it is necessary to improve the language of your manuscript to an acceptable level.

We look forward to receiving your revised manuscript.

Kind regards,

Reza Yousefi, Ph.D

Academic Editor

PLOS ONE

Journal Requirements:

"This work was supported by the research council of Tarbiat Modares University and Ministry of Sciences, Researches, and Technology, Iran."

"NO authors have competing interests"

We note that one or more of the authors are employed by a commercial company: Mah Behin Gene Gostaran Company.

4.1. Please provide an amended Funding Statement declaring this commercial affiliation, as well as a statement regarding the Role of Funders in your study. If the funding organization did not play a role in the study design, data collection and analysis, decision to publish, or preparation of the manuscript and only provided financial support in the form of authors' salaries and/or research materials, please review your statements relating to the author contributions, and ensure you have specifically and accurately indicated the role(s) that these authors had in your study. You can update author roles in the Author Contributions section of the online submission form.

4.2. Please also provide an updated Competing Interests Statement declaring this commercial affiliation along with any other relevant declarations relating to employment, consultancy, patents, products in development, or marketed products, etc. 

5. Please ensure that you refer to Figure 'Graphical Abstract' in your text as, if accepted, production will need this reference to link the reader to the figure.

Reviewers' comments:

Reviewer's Responses to Questions

**Comments to the Author**

1. Is the manuscript technically sound, and do the data support the conclusions?

Reviewer #1: No

Reviewer #2: Yes

2. Has the statistical analysis been performed appropriately and rigorously? 

Reviewer #1: No

Reviewer #2: N/A

3. Have the authors made all data underlying the findings in their manuscript fully available?

Reviewer #1: No

Reviewer #2: No

4. Is the manuscript presented in an intelligible fashion and written in standard English?

Reviewer #1: No

Reviewer #2: No

5. Review Comments to the Author

Reviewer #1: Sajed et al. present results on the effect of heat and H2O2 on artemin, a protein from Artemia involved in survival of neurons. Despite the large number of experimental tools, there is a lack of experimental and interpretative rigor in the techniques used. For these reasons, some of which are detailed below, the conclusions reported in the text are not supported by the experimental results. In addition to these observations, the novelty of the work is not clear. What is really the applied outcome of the work? The current work has added additional information to artemin knowledge or not.

In conclusion, it is not possible to recommend acceptance of the manuscript.

Majors:

-What is the quality/state of the produced protein? Is the protein folded (include circular dichroism spectra)? Is the protein aggregated (include gel filtration information)? Is there a SDS-PAGE of the preparation that can be evaluated? The second (Fig 2A) and seventh (Fig. 2B) lanes are likely to refer to the produced protein. It seems to show the presence of monomer, dimer and a very large aggregate. Is it possible to separate those species by gel filtration chromatography? In this case, are the species states reversible? Why are the bands at very high concentration SDS-resistant?

-Have the authors considered aggregation/precipitation as a function of temperature-induced unfolding-refolding, and also after H2O2 treatment? See for instance: 1)Fig2A: samples at 70 and 80C have much less protein than the other lanes (authors need to use a software tool such as ImageJ to evaluate protein quantity in each lane). 2)Fig.2B: same ´protein missing problem´; samples DTT+GA- and DTT-GA+ have much less proteins than the others. 3)Figs 5 and 7 have the same problem. Same ´protein missing problem´ in many lanes. 4)Fig.3. Distributions ate 70 and 80C are so thin that it may indicate protein precipitated that was not recovered. This is a very important issue because it may explain the decrease in fluorescence in Figs1A and 4A (protein precipitated that was not recovered).

-Fluorescence. 1)An amino acid sequence should be added to help quantify the amount of Tyr and Trp. Authors state that Fig1A refers to ´Trp fluorescence emission´ but Tyr residues, if present, also contribute to emitted fluorescence. 2)The intensity of Trp fluorescence is temperature-dependent. Any study of heat-induced unfolding followed by Trp fluorescence has to present a curve control using N-acetyl-tryptophanamide (NATA) at the same buffer (there are plenty of book chapter and reviews about the topic). The authors need to comment on that. 3)Figs. 4A and 6A: Trp fluorescence is quenched by hydrogen peroxide. Is there a control to the experiments? 5)What is the rational for the bis-ANS concentration used? Is there a bis-ANS titration experiment to be evaluated?

-SDS-PAGE figures are of low quality and have little information about protein quantity. See that even the loaded marker has a band of very high molecular mass (Fig. 7 is the worse). Additionally, authors need to use a software tool such as ImageJ to evaluate protein quantity in each lane.

-Chaperone activity. Data seem to have some potential but it measures no refolding, maybe it probes protection against aggregation. To probe refolding, authors need to show that lysozyme regain its folded state (CD, NMR, enzymology, other). But even protection against aggregation seem to be minor, 20% or less.

Minors:

-Protein concentration should be given in micromolar to facilitate understanding of the amount of additives used.

-Material and Methods description lack some information, as for instance the temperature of some experiments.

Reviewer #2: This study examined the effects of temperature and an oxidizing agent (hydrogen peroxide) on the structure and chaperoning capacity of the brine shrimp protein artemin. This molecular chaperone has received a great deal of study in the past, and the present work is thus built on a large foundation of information. The principal contribution of these new data is the demonstration of how changes in oligomerization of artemin are driven by temperature and oxidative state, and how these structural changes lead to influences on chaperone potential. Activation of artemin relies chiefly on temperature-induced alterations in conformation that favor dimerization of the protein. Disulfide bridges between artemin monomers are shown to play important roles in the dimerization process. The work involves a large and diverse set of experiments, but the conclusions are nicely summarized in the two “schemes” the authors provide.

The experimental work appears to have been done carefully and the interpretations of the findings are well-supported by the results. The writing is generally clear, but there are dozens of errors in syntax and grammar that need to be corrected.

My main criticism of the paper is that the authors make no effort to link these in vitro findings to the biology of the organism itself. For example, they provide no information on the temperatures that the cysts of the shrimp experience under natural field conditions. Thus, it is not possible to determine whether the experimental conditions used with the isolated protein resemble the thermal conditions found in nature. I am concerned that some of the effects on the protein that were manifested under extremes of stress, notably some of the highest experimental temperatures, would never occur in the actual organism. In summary, whereas the in vitro biochemistry here is of high quality and yields interesting results, there is no way of determining whether these results, notably the stress-driven changes in dimerization state, actually occur in nature. The authors should try to link their in vitro findings to in vivo phenomena.

6. PLOS authors have the option to publish the peer review history of their article (what does this mean?). If published, this will include your full peer review and any attached files.

Reviewer #1: No

Reviewer #2: No

---

## [Author Response · Author response to Decision Letter 0]

16 Sep 2020

Dear Editors-in-Chief, 

We are grateful for your consideration of this manuscript (Manuscript ID: PONE-D-20-06479), and we also very much appreciate your suggestions, which have been very helpful in improving the manuscript. We also thank the reviewers for their careful reading of our text. All the comments we received on this study have been taken into account in improving the quality of the article, and we present our reply to each of them separately. All corrections and modifications are highlighted in red in the revised version of the manuscript.

We hope that these changes to the manuscript will facilitate the decision to publish this study in your journal. We have made a considerable effort to take into account the interesting suggestions proposed by the reviewers. In any case, we are open to consideration of any further comment on our answers.

Sincerely,

Reza H Sajedi, PhD

Professor of Biochemistry

Department of Biochemistry,

Faculty of Biological Sciences,

Tarbiat Modares University, Tehran, Iran

e.mail: sajedi_r@modares.ac.ir

 

Comments from reviewers:

Reviewer 1:

Sajedi et al. present results on the effect of heat and H2O2 on artemin, a protein from Artemia involved in survival of neurons. Despite the large number of experimental tools, there is a lack of experimental and interpretative rigor in the techniques used. For these reasons, some of which are detailed below, the conclusions reported in the text are not supported by the experimental results. In addition to these observations, the novelty of the work is not clear. What is really the applied outcome of the work? The current work has added additional information to artemin knowledge or not. In conclusion, it is not possible to recommend acceptance of the manuscript.

ANSWER: Thank you very much for sharing us your voluble comments and concerns. As the first point, it is needed to mention that artemin is a thermostable protein in Artemia embryos (the name of the protein is similar to a member of the glial cell-line derived neurotrophic factor family, expressing in numerous human mammary carcinoma cell lines). Artemin acts as a highly efficient molecular chaperone in the encysted Artemia embryos (brine shrimps) against extreme environmental stress conditions. We previously worked on the chaperoning potency of this protein and confirmed that artemin can act as an efficient molecular chaperone in vitro and in vivo (References 5, 8, 11-13, 15 in the manuscript). Moreover, we used recombinant artemin for different purposes. For example, enhancing the soluble production of some aggregation-prone proteins in bacterial cells such as aequorin (Khosrowabadi et al., 2018), luciferase (Takalloo et al., 2016), endostatin (non-published data), and also increasing the thermostability of an industrial enzyme (xylanase) (non-published yet). Moreover, the anti-amyloidogenic effect of artemin on α-synuclein protein (Marvastizadeh et al., 2020) and blocking fibrillization of tau protein by artemin (Khosravi et al., 2017) were further confirmed. More recently, artemin gene was transferred into plant cells and expressed, in order to produce the stress-tolerant transgenic plants and the first generation of the plants showed a higher resistance against desiccation and high temperatures (data not published yet).

Despite such successes, there were no enough details on the activation mechanisms or structural modifications of this chaperone. In addition, the structural changes of artemin under oxidative stress has not been investigated to date. Accordingly, in the present study, we tried to provide some basis on the induced conformational changes of artemin upon exposing to stress conditions and the mechanisms involved in interactions with the protein substrates. This is the first report in providing details on the conformational changes of artemin’s subunits (in particular during oxidative stress). We believe that these information would help us not only to justify the protective function of artemin in Artemia cysts or other transgenic organisms, but also to effectively select more target protein substrates for further in vitro and in vivo studies. 

According to the respectful reviewer’s comments, we have expanded details in the introduction and discussion to more clarify the aim and novelty of the present work for the reviewer as the other readers (Page 4; Lines 6-11, 14-16, 20-22, Page 18; Lines 13-16). In addition, we added some related explanations (as also asked by the respectful reviewer 2) in the conclusion (Page 28; Lines 19-22, Page 29; Lines 1-18, 22, 23, Page 30; Line 1).

 Major comments: 

1. What is the quality/state of the produced protein? Is the protein folded (include circular dichroism spectra)? Is the protein aggregated (include gel filtration information)? Is there a SDS-PAGE of the preparation that can be evaluated? The second (Fig 2A) and seventh (Fig. 2B) lanes are likely to refer to the produced protein. It seems to show the presence of monomer, dimer and a very large aggregate. Is it possible to separate those species by gel filtration chromatography? In this case, are the species states reversible? Why are the bands at very high concentration SDS-resistant? 

ANSWER: Regarding the state of the protein, artemin (at 25 ºC) has a normal three dimensional (3D) structure, as also confirmed by fluorescence spectroscopy (Fig. 1A, Fig. 4A). We also checked the secondary structure of purified artemin by CD spectroscopy (shown in below). Secondary structure analysis by K2D2 online software predicted an α-helical content of approximately 47.89%, followed by random coil (41.91%), and β-sheet (10.28%), as also showed in our previous works. 

Moreover, regarding the form of artemin in normal conditions, please notice that, like other small heat shock proteins (such as Hsp26, Hsp20.1 and Hsp14.1), artemin probably exists in a monomer–oligomer equilibrium in the cells (Scheme 1, Page 29; Lines 19-20). Overall, dynamic oligomerization and the ability to switch between oligomeric states appear to be crucial for the function of a variety of known chaperones, but in very different ways. Previous SEM analysis also revealed the Rosset like structure of artemin composing hexameric structure of artemin in the Artemia (franciscana) cysts (Graaf et al., 1990; Chen et al., 2007). It is suggested that environmental stresses and/or intracellular portion of proteins may shift the equilibrium towards the active dimer forms through forming disulfide bridges between the monomers. Our cross-link analysis by SDS-PAGE revealed that the ratio of dimeric states of the chaperone was changed upon exposing to stress conditions (Fig. 2 A, Fig. 5, Fig. 7). 

Analysis of purified artemin by SDS-PAGE (in below) under reducing and non-reducing conditions was performed, which revealed monomeric forms (under reducing), and dimeric/other oligomeric structures (under non-reducing conditions).

Purified artemin fractions in non-reduced (1) and reduced (2-11) SDS-PAGE.

In addition, performing gel filtration needs a high concentration of purified protein. Gel filtration can dilute the protein concentration 10 times. Thus, if our protein concentration is not high enough, the monitor can not detect all oligomeric forms of the protein after gel filtration. Accordingly, we preferred to use cross-link analysis followed by SDS-PAGE. In comparison, using SDS-PAGE followed by staining with nitrate silver enabled us to use lower concentrations of proteins and also detect various forms of protein even at very low concentrations (~5 ng). 

Regarding the reversibility of the protein states, our suggestion is that under normal (or non-stress) conditions, some transitions may occur between different protein states. However, in stress conditions, we checked the reversibility of artemin modifications after exposing to stress conditions (Fig. 8), and the results of fluorescence spectroscopy indicated that the structural changes were irreversible. 

About the final question, it is needed to mention that the protein samples were treated with 0.5% glutaraldehyde as a fixative agent and then loaded onto the 10% SDS-PAGE. The fixed protein subunits by glutaraldehyde prevented the subunits dissociation upon exposing to SDS or other agents during SDS-PAGE analysis. 

2. Have the authors considered aggregation/precipitation as a function of temperature-induced unfolding-refolding, and also after H2O2 treatment? See for instance: 1)Fig2A: samples at 70 and 80C have much less protein than the other lanes (authors need to use a software tool such as ImageJ to evaluate protein quantity in each lane). 2)Fig.2B: same ´protein missing problem´; samples DTT+GA- and DTT-GA+ have much less proteins than the others. 3)Figs 5 and 7 have the same problem. Same ´protein missing problem´ in many lanes. 4)Fig.3. Distributions at 70 and 80C are so thin that it may indicate protein precipitated that was not recovered. This is a very important issue because it may explain the decrease in fluorescence in Figs1A and 4A (protein precipitated that was not recovered).

ANSWER: We completely agree with the respectful reviewer’s comment. We also believe that exposing the chaperone to elevated temperatures (≥ 70 ºC), as well as higher concentrations of H2O2 (≥ 80 mM) probably lead to forming higher molecular weights of protein assembles and probably protein aggregates as we also mentioned in our manuscript before (Page 15; lines 11-13, Page 15; lines 17-20, Page 18; lines 5-7; Page 19; lines 14-18 in previous version of the manuscript). As the dear reviewer also mentioned, these aggregates or precipitates could also result in decrease in the fluorescence intensity as was observed for heated artemin at elevated temperatures or oxidized protein with higher concentrations of the oxidant (Fig. 1A, Fig. 3A). We referred these suggestions in the previous version of the manuscript, but we also proposed that the presence of protein substrates may stabilize artemin conformations at higher temperatures or oxidant contents and protect it from self-association/precipitation and this may even induce the reversibility of structural changes of artemin upon exposing to stress conditions, however in the present study we did not test this hypothesis (Page 16; lines 8-10, Page 21; lines 7-11 in previous version of the manuscript). 

Besides, we analyzed the percent of dimeric forms of the treated proteins (as the active state of the chaperone) using ImageJ software and added related data in the manuscript (Page 7; Lines 16-18, Page 10; Line 8-9, 17-18, Page 11; Lines 2-3, Page 13; Lines 1, 6-7, Page 14; Lines 10-11, 14-15, Page 21; Lines 1-2, Page 23; Line 22, Figures 2B, D, 5B, 7B).

3. Fluorescence. 1) An amino acid sequence should be added to help quantify the amount of Tyr and Trp. Authors state that Fig1A refers to ´Trp fluorescence emission´ but Tyr residues, if present, also contribute to emitted fluorescence. 2)The intensity of Trp fluorescence is temperature-dependent. Any study of heat-induced unfolding followed by Trp fluorescence has to present a curve control using N-acetyl-tryptophanamide (NATA) at the same buffer (there are plenty of book chapter and reviews about the topic). The authors need to comment on that. 3)Figs. 4A and 6A: Trp fluorescence is quenched by hydrogen peroxide. Is there a control to the experiments? 5)What is the rational for the bis-ANS concentration used? Is there a bis-ANS titration experiment to be evaluated?

ANSWER: 1) According to the respectful reviewer’s comment, the amino acid sequences of the protein was added to the manuscript (Page 19; Line 17, Page 31; F1 S1).

Regarding the second comment, although tyrosine (Tyr) has a quantum yield similar to Trp, the indole group of Trp is considered the dominant source of UV absorbance at ∼280 nm and emission at ∼350 nm in proteins (Teale et al., 1957). Furthermore, in native proteins, Tyr emission is often quenched, presumably by its interaction with the peptide chain or via energy transfer to Trp (Lakowicz et al., 2006). In comparison, the spectroscopic properties of Trp are complex, in particular the high sensitivity to the (local) environment and its (at least) two different fluorescence lifetimes (∼0.5 and ∼3.1 ns) (Amar et al., 2014). Accordingly, the fluorescence of the protein upon excitation at 280 nm generally referred as the Trp fluorescence. But as the reviewers suggested, we corrected the “Trp fluorescence” phrases to avoid any misunderstanding (Page 9; Line 18, Page 12; Line 13).

2) About using a curve control using NATA; the quantum yield of fluorescence of the Trp residues generally diminished as the temperature increased, also with no change in the shape of their emission spectra (Gally and Edelman, 1962). Besides, NATA is often used as a model for the Trp residues in proteins and peptides. It is used to quantify the number of Trp residues changed in proteins. However, here we tried to compare the conformational changes between different treated proteins, and not to quantify the number of exposed Trp residues. There are also many reliable studies, in which the comparison has been performed in the similar manner as we used (Wang et al., 2002; Gosslau et al., 2001; Waseem Akhtar et al., 2004; Choi et al., 2013; Shearstone and Baneyx, 1999; Dahl eta al., 2015; Jung et al., 2015; Fan et al., 2006 and so on). 

3, 4) Yes, we are agreed with the respectful reviewer that the quenching of Trp and Tyr fluorescence upon exposing to the oxidant may also occur as a result of oxidation of the aromatic residues. However, in this regard, the accessibility of the fluorophore to the solvent and thus to the quencher depends on its position within the protein. Trp is quite sensitive to its local environment. Then, buried Trp residues should have a lower accessibility to the solvent that those present at the surface (J. R. Albani, Elsevier Science, 2011). Accordingly, the residues on the surface of the protein will be influenced to a greater degree by the oxidant, while the buried Trp residues are not easily accessible to the solvent. There are also many reports in which the fluorescence intensity of a protein has been measured in the similar manner as we used (Wang et al., 2002; Khoshaman et al., 2015; Gosslau et al., 2001, and so on). As control, we added H2O2 at different concentrations to the protein solutions and recorded the fluorescence intensity after 2 minutes, and the differences between the control (0 mM H2O2) and the samples with higher oxidant contents were insignificant. Accordingly, as another probable reason, the decrease in the fluorescence intensity of the oxidized protein may be as a result of the formed disulﬁde bonds in the oxidized protein. The prone contacts between cystines and the aromatic rings lead to the ﬂuorescence decay. We also added the earlier possibility into the manuscript for more clarifications (Page 23; Lines 8-10).

5) ANS generally is used in a broad range of 10-100 µM, but we chose the optimum concentration (30 µM) based on a dozen of studies reported previously (Das et al., 1995; Wang et al., 2002; Gosslau et al., 2001; Jung et al., 2015). 

4. SDS-PAGE figures are of low quality and have little information about protein quantity. See that even the loaded marker has a band of very high molecular mass (Fig. 7 is the worse). Additionally, authors need to use a software tool such as ImageJ to evaluate protein quantity in each lane. 

ANSWER: The figures were provided in a very high quality version (in TIFF format with dpi: 300). The quality probably decreased during converting TIFF to PDF by the journal. We have provided some of the figure (in DOX version) for more clarifications (Please check the below figures for example). 

 

Figure 2

 

Figure 7

Moreover, as the dear reviewer suggested, we evaluated the quantity of dimeric forms (as the active state of the chaperone) in each lane by ImageJ software (Page 7; Lines 16-18, Page 10; Line 8-9, 17-18, Page 11; Lines 2-3, Page 13; Lines 1, 6-7, Page 14; Lines 10-11, 14-15, Page 21; Lines 1-2, Page 23; Line 22, Figures 2B, D, 5B, 7B).

5. Chaperone activity. Data seem to have some potential but it measures no refolding, maybe it probes protection against aggregation. To probe refolding, authors need to show that lysozyme regain its folded state (CD, NMR, enzymology, other). But even protection against aggregation seem to be minor, 20% or less.

ANSWER: Thank you for this point. Please notice that it is not possible to determine the CD or NMR spectra of a protein when we want to examine the interactions between two or more proteins. In the case of molecular chaperones, refolding experiments are among the most popular and efficient methods used for evaluating the potential of molecular chaperones as also used for many other chaperones (Wang et al., 2010; Dong et al., 2002; Choi et al., 2013; Rozema et al., 1996; Wang et al., 2010; Dong et al., 2004; Dong et al., 2001; Gull et al., 2017; Takalloo et al., 2019; Antonio-Pérez et al., 2012; Niknaddaf, 2018). But regarding the refolding yield, depending on type of the target protein substrate and the protein substrate: chaperone ratio, results may be different. Finally, please remind that the main purpose of performing refolding experiments was not increasing the refolded yield of lysozyme, but simply compare the chaperonin potency of different treated artemin (heated, oxidized, and heated oxidized artemin).

Minor comments:

1. Protein concentration should be given in micromolar to facilitate understanding of the amount of additives used. 

ANSWER: As the respected reviewer suggested, we replaced all protein concentrations in mg/mL with µM (Page 6; Line 19, Page 7; Lines 3, 7, 10, 13, 20, Page 8; Lines 4, 19, Page 9; Line 19, Page 10; Lines 2, 18, 22, Page 12; Lines 12, 16, Page 13; Line 7, Page 14; Lines 2, 7, 15, Page 15; Lines 9, 13, Page 16; Line 13). 

2. Material and Methods description lack some information, as for instance the temperature of some experiments. 

ANSWER: We added more details regarding the experimental preparation to the manuscript (Page 6; Lines 1-10, 14, 15, Page 7; Lines 1, 15).

Apart from the changes (requested by respected reviewers), some other alterations have been made and highlighted in red in the revised manuscript, e.g.: Page 2; Lines 2, 6, 10, 12, 18, Page 3; Lines 8, 9, 11, 18, 21, 23, Page 4; Lines 4, 5, 12, 13, 17, Page 7; Line 4, 6, 11, 12, 19, 21, Page 8; Lines 2, 6, 11, 17, 20, Page 9; Lines 1, 2, 20, Page 10; Lines 3, 10, 19, Page 11; Lines 7, 8, 18, Page 12; Line 6, Page 14; Lines 4, 9, 17, Page 15; Lines 3, 21, Page 16; Lines 4, 5, 6, Page 17; Lines 1, 2, 10, 12, Page 18; Lines 3, 5, 7, 8, 9, 12, Page 19; Lines 2, 3, 5, 6, 7, 8, 9, 1, 11, 15, 16, 17, 18, 19, Page 20; Lines 3, 5, 6, 7, 9, 10, 11, 12, 13, 14, 15, 16, 17, 18, 19, 21, 23, Page 21; Lines 4, 6, 8, 9, 10, 12, 13, 14, 15, 16, 17, 19, 20, 22, 23, Page 22; Lines 5, 8, 10, 12, 13, 14, 15, 16, 20, 22, 23, Page 23; Lines 1, 2, 3, 4, 5, 6, 14, 15, 17, 18, 20, 21, 22, 23, Page 24; Lines 6, 9, 11, 12, 13, 14, 15, 17, 18, 19, 20, Page 25; Lines 2, 3, 6, 9, 10, 11, 12, 13, 14, 15, 16, 17, 19, 20, 22, Page 26; Line 2, 4, 8, Page 28; Line 6, Page 29; Line 19.

Due to the alterations (requested by respected reviewers) 15 new references (Ref 1, 2, 16, 17, 18, 20, 21, 41, 47-53) were added to the manuscript.

Again, we appreciate all of your insightful comments. We worked hard to be responsive to them. Thank you for taking the time and energy to help us improve the paper. 

 

-Reviewer 2

This study examined the effects of temperature and an oxidizing agent (hydrogen peroxide) on the structure and chaperoning capacity of the brine shrimp protein artemin. This molecular chaperone has received a great deal of study in the past, and the present work is thus built on a large foundation of information. The principal contribution of these new data is the demonstration of how changes in oligomerization of artemin are driven by temperature and oxidative state, and how these structural changes lead to influences on chaperone potential. Activation of artemin relies chiefly on temperature-induced alterations in conformation that favor dimerization of the protein. Disulfide bridges between artemin monomers are shown to play important roles in the dimerization process. The work involves a large and diverse set of experiments, but the conclusions are nicely summarized in the two “schemes” the authors provide.

The experimental work appears to have been done carefully and the interpretations of the findings are well-supported by the results. The writing is generally clear, but there are dozens of errors in syntax and grammar that need to be corrected.

My main criticism of the paper is that the authors make no effort to link these in vitro findings to the biology of the organism itself. For example, they provide no information on the temperatures that the cysts of the shrimp experience under natural field conditions. Thus, it is not possible to determine whether the experimental conditions used with the isolated protein resemble the thermal conditions found in nature. I am concerned that some of the effects on the protein that were manifested under extremes of stress, notably some of the highest experimental temperatures, would never occur in the actual organism. In summary, whereas the in vitro biochemistry here is of high quality and yields interesting results, there is no way of determining whether these results, notably the stress-driven changes in dimerization state, actually occur in nature. The authors should try to link their in vitro findings to in vivo phenomena. 

ANSWER: The constructive comments by the respectful reviewer are really appreciated. Accordingly, we have carefully revised the conclusion and other sections of the manuscript and added some related descriptions regarding the biology, lifetime and the living environments of the organism and tried to explore the relationship between our data and Artemia biology (Page 3; Lines 2-6, Page 28; Lines 19-22, Page 29; Lines 1-18, 22-23, Page 30; Line 1). 

For more clarification, it is needed to mention that our main purpose was not solely to link the chaperoning function of artemin to its conformational changes during stress condition in Artemia cysts as its origin. To date, we have used recombinant artemin for different purposes. For example, enhancing the soluble production of some aggregation-prone proteins in bacterial cells such as aequorin (Khosrowabadi et al., 2018), luceiferase (Takalloo et al., 2016), endostatin (non-published data), and also increasing the thermostability of an industrial enzyme (xylanase) (non-published yet). Moreover, the anti-amyloidogenic effect of artemin on α-synuclein protein (Marvastizadeh et al., 2020) and blocking fibrillization of tau protein by artemin (Khosravi et al., 2017) were further confirmed. More recently, artemin gene was transferred into plant cells and expressed, in order to produce the stress-tolerant transgenic plants and the first generation of the plants showed a higher resistance against desiccation and high temperatures (data not published yet).

Despite such successes, there were no enough details on the activation mechanisms or structural modifications of this chaperone. In addition, the structural changes of artemin under oxidative stress has not been investigated to date. Accordingly, in the present study, we tried to provide some basis on the induced conformational changes of artemin upon exposing to stress conditions and the mechanisms involved in interactions with the protein substrates. This is the first report in providing details on the conformational changes of artemin’s subunits (in particular during oxidative stress). We believe that these information would help us not only to justify the protective function of artemin in Artemia cysts or other transgenic organisms, but also to effectively select more target protein substrates for further in vitro and in vivo studies. 

Accordingly, to avoid any misunderstanding, we have expanded details in the introduction and discussion to more clarify the aim and novelty of the present work for the reviewer as the other readers (Page 4; Lines 6-11, 14-16, 20-22, Page 18; Lines 13-16).

Besides, we also re-read the whole manuscript and corrected the grammatical errors and typing mistakes as much as possibl, e.g.: Page 2; Lines 2, 6, 10, 12, 18, Page 3; Lines 8, 9, 11, 18, 21, 23, Page 4; Lines 4, 5, 12, 13, 17, Page 7; Line 4, 6, 11, 12, 19, 21, Page 8; Lines 2, 6, 11, 17, 20, Page 9; Lines 1, 2, 20, Page 10; Lines 3, 10, 19, Page 11; Lines 7, 8, 18, Page 12; Line 6, Page 14; Lines 4, 9, 17, Page 15; Lines 3, 21, Page 16; Lines 4, 5, 6, Page 17; Lines 1, 2, 10, 12, Page 18; Lines 3, 5, 7, 8, 9, 12, Page 19; Lines 2, 3, 5, 6, 7, 8, 9, 1, 11, 15, 16, 17, 18, 19, Page 20; Lines 3, 5, 6, 7, 9, 10, 11, 12, 13, 14, 15, 16, 17, 18, 19, 21, 23, Page 21; Lines 4, 6, 8, 9, 10, 12, 13, 14, 15, 16, 17, 19, 20, 22, 23, Page 22; Lines 5, 8, 10, 12, 13, 14, 15, 16, 20, 22, 23, Page 23; Lines 1, 2, 3, 4, 5, 6, 14, 15, 17, 18, 20, 21, 22, 23, Page 24; Lines 6, 9, 11, 12, 13, 14, 15, 17, 18, 19, 20, Page 25; Lines 2, 3, 6, 9, 10, 11, 12, 13, 14, 15, 16, 17, 19, 20, 22, Page 26; Line 2, 4, 8, Page 28; Line 6, Page 29; Line 19.

Due to the alterations (requested by respected reviewers) 15 new references (Ref 1, 2, 16, 17, 18, 20, 21, 41, 47-53) were added to the manuscript.

Again, we appreciate you taking the time to offer us your comments and insights related to the paper.

---

## [Decision Letter · Decision Letter 1]

19 Oct 2020

PONE-D-20-06479R1

Stress-dependent conformational changes of artemin: eﬀects of heat and oxidant

PLOS ONE

Dear Dr. Sajedi,

Thank you for submitting your manuscript to PLOS ONE. After careful consideration, we feel that it has merit but does not fully meet PLOS ONE’s publication criteria as it currently stands. Therefore, we invite you to submit a revised version of the manuscript that addresses the points raised during the review process.

Your article will be accepted if the following changes are applied correctly.

1. The numbers identifying the affiliations of authors are not used correctly.

2. The language of this manuscript still needs significant improvement. Below please find some examples of typo grammatical errors which must be fixed.

Abstract section, lines 3-6, the sentence needs amending.

Pg.4, line 2: moderates should be replaced with modulates.

Pg.6, line 5: Microtubule must be corrected as micro tube.

3. Length of the final conclusion needs to be greatly shortened and additional and repetitive explanations must to be avoided.

4. While drawing the GA and Scheme 1, if more than one color is used, it will probably be easier for readers to understand the messages.

.==============================

We look forward to receiving your revised manuscript.

Kind regards,

Reza Yousefi, Ph.D

Academic Editor

PLOS ONE

Additional Editor Comments (if provided):

Your article will be accepted if the following changes are applied correctly.

1) The numbers identifying the affiliations of authors are not used correctly.

2) The language of this manuscript still needs significant improvement. Below please find some examples of typo grammatical errors which must be fixed.

Abstract section, lines 3-6, the sentence needs amending.

Pg.4, line 2: moderates should be replaced with modulates.

Pg.6, line 5: Microtubule must be corrected as micro tube.

3) Length of the final conclusion needs to be greatly shortened and additional and repetitive explanations must to be avoided.

4) While drawing the GA and Scheme 1, if more than one color is used, it will probably be easier for readers to understand their messages.

Reviewers' comments:

Reviewer's Responses to Questions

**Comments to the Author**

1. If the authors have adequately addressed your comments raised in a previous round of review and you feel that this manuscript is now acceptable for publication, you may indicate that here to bypass the “Comments to the Author” section, enter your conflict of interest statement in the “Confidential to Editor” section, and submit your "Accept" recommendation.

Reviewer #3: All comments have been addressed

2. Is the manuscript technically sound, and do the data support the conclusions?

Reviewer #3: Yes

3. Has the statistical analysis been performed appropriately and rigorously? 

Reviewer #3: Yes

4. Have the authors made all data underlying the findings in their manuscript fully available?

Reviewer #3: Yes

5. Is the manuscript presented in an intelligible fashion and written in standard English?

Reviewer #3: Yes

6. Review Comments to the Author

Reviewer #3: No comment. In my opinion, all of the points raised in the first revision were addressed by the authors. Accordingly, the manuscript is acceptable for publication in PLOS ONE.

7. PLOS authors have the option to publish the peer review history of their article (what does this mean?). If published, this will include your full peer review and any attached files.

Reviewer #3: **Yes: **Mohsen Asghari

---

## [Author Response · Author response to Decision Letter 1]

25 Oct 2020

Dear Editors-in-Chief, 

We are grateful for your consideration of this manuscript (Manuscript ID: PONE-D-20-06479). All the comments we received on this study have been taken into account in improving the quality of the article, and we present our reply to each of them separately. All corrections and modifications are highlighted in red in the revised version of the manuscript.

1. The numbers identifying the affiliations of authors were checked and corrected (Page 1).

2. We re-read the whole manuscript and corrected the grammatical errors and typing mistakes as much as possible, e.g.: Page 2; Lines 4, 5, 6, 7, 10, 14, Page 3; Line 11, Page 4; Lines 1, 3, 20, 21, Page 6; Lines 5, 6, 8, Page 9; Lines 9, 10, Page 10; Lines 1, 9, 16, Page 11; Lines 3, 4, 6, 21, Page 12; Line 11, Page 16; Lines 10, 16, 17, Page 17; Lines 1, 3, 14, 19, 20, Page 18; Lines 9, 10, 13, 14, Page 19; Lines , 18, 20, Page 20; Lines 3, 4, 14, Page 21; Lines 5, 15, 18, 20, 22, 23, Page 22; Lines 1, 3, 6, 9, 14, 21, Page 23; Lines 1, 9, 10, Page 24; Lines 17, 18, 22, Page 25; Line 18, Page 26; Line 9, Page 27; Lines 5, 19, 20, 23.

3. Regarding the conclusion section, as the respectful reviewer recommended us to add some related information about the physiology of Artemia cysts in the conclusion section, and accordingly the conclusion was too lengthy. As the dear editor suggested, we removed the first part of the conclusion and added that to the discussion (Page 28; Lines 3-18), and shortened conclusion as much as possible (Page 28; Lines 21-22, Page 29; Lines 1-6).

4. We corrected the scheme 1 and graphical abstract as the respectful reviewer suggested.

Again, we appreciate you taking the time to offer us your comments and insights related to the paper. We hope that these changes to the manuscript will facilitate the decision to publish this study in your journal. In any case, we are open to consideration of any further comment on our manuscript.

Sincerely,

Reza H Sajedi, PhD

Professor of Biochemistry

Department of Biochemistry,

Faculty of Biological Sciences,

Tarbiat Modares University, Tehran, Iran

E.mail: sajedi_r@modares.ac.ir

---

## [Editor Report · Decision Letter 2]

29 Oct 2020

Stress-dependent conformational changes of artemin: eﬀects of heat and oxidant

PONE-D-20-06479R2

Dear Professor Sajedi,

We’re pleased to inform you that your manuscript has been judged scientifically suitable for publication and will be formally accepted for publication once it meets all outstanding technical requirements.

Kind regards,

Reza Yousefi, Ph.D

Academic Editor

PLOS ONE
---

## [Editor Report · Acceptance letter]

6 Nov 2020

PONE-D-20-06479R2 

Stress-dependent conformational changes of artemin: eﬀects of heat and oxidant 

Dear Dr. Sajedi:

I'm pleased to inform you that your manuscript has been deemed suitable for publication in PLOS ONE. Congratulations! Your manuscript is now with our production department. 

Kind regards, 

on behalf of

Professor Reza Yousefi 

Academic Editor

PLOS ONE